# Building Blocks for Robust and Effective Semi-Supervised Real-World Object Detection

**Moussa Kassem Sbeyti**[1,2,3]   **Nadja Klein**[1]
**Azarm Nowzad**[3]   **Fikret Sivrikaya**[2]   **Sahin Albayrak**[2]

[1]**Scientific Computing Center, Karlsruhe Institute of Technology**
[2]**DAI-Labor, Technische Universität Berlin**   [3]**Continental AG**
{moussa.sbeyti, nadja.klein}@kit.edu
azarm.nowzad@continental-corporation.com
{fikret.sivrikaya, sahin.albayrak}@dai-labor.de

**Reviewed on OpenReview:** https://openreview.net/forum?id=vRYt8QLKqK

## Abstract

Semi-supervised object detection (SSOD) based on pseudo-labeling significantly reduces dependence on large labeled datasets by effectively leveraging both labeled and unlabeled data. However, real-world applications of SSOD often face critical challenges, including class imbalance, label noise, and labeling errors. We present an in-depth analysis of SSOD under real-world conditions, uncovering causes of suboptimal pseudo-labeling and key trade-offs between label quality and quantity. Based on our findings, we propose four building blocks that can be seamlessly integrated into an SSOD framework. **Rare Class Collage (RCC):** a data augmentation method that enhances the representation of rare classes by creating collages of rare objects. **Rare Class Focus (RCF):** a stratified batch sampling strategy that ensures a more balanced representation of all classes during training. **Ground Truth Label Correction (GLC):** a label refinement method that identifies and corrects false, missing, and noisy ground truth labels by leveraging the consistency of teacher model predictions. **Pseudo-Label Selection (PLS):** a selection method for removing low-quality pseudo-labeled images, guided by a novel metric estimating the missing detection rate while accounting for class rarity. We validate our methods through comprehensive experiments on autonomous driving datasets, resulting in up to 6% increase in SSOD performance. Overall, our investigation and novel, data-centric, and broadly applicable building blocks enable robust and effective SSOD in complex, real-world scenarios. Code is available at https://mos-ks.github.io/publications.

## 1 Introduction

Training robust object detectors for real-world applications is challenging due to the high cost of obtaining large labeled datasets with accurate labels. Semi-supervised learning (SSL) offers a promising solution by leveraging both labeled and unlabeled data. Existing SSL methods often utilize a teacher-student framework, where a teacher model trained on labeled data generates pseudo-labels for unlabeled data, which are then used together to train a student model (Li et al., 2020b; Sohn et al., 2020; Yang et al., 2021; Xu et al., 2021; Liu et al., 2021; Zhang et al., 2022; Liu et al., 2022b; Mi et al., 2022; Li et al., 2022c). While effective, the potential of SSL is often hindered by the quality of both labeled and pseudo-labeled data.

**Acknowledgment.** This work has been funded by the KiKIT (The Pilot Program for Core-Informatics at the KIT) of the Helmholtz Association. It was also supported by the Helmholtz Association Initiative and Networking Fund on the HAICORE@KIT partition. Additionally, this work acknowledges financial support by the German Federal Ministry for Economic Affairs and Climate Action (BMWK) and the European Union within the project "just better DATA (jbDATA)", grant no. 19A23003A.

We identify three primary challenges that affect label quality and, consequently, the effectiveness of pseudo-labeling-based SSL frameworks. These are: (1) class imbalance in both labeled and pseudo-labeled data, which leads to poor generalization on underrepresented classes; (2) noise in ground truth labels, including incorrect or missing labels, which compromises training; and (3) incomplete or inaccurate pseudo-labels generated by the teacher, particularly for underrepresented classes, which propagate errors to the student. To address these challenges, we propose the following four novel building blocks aimed at increasing the effectiveness and robustness of SSOD by enhancing label quality, utilization, and class balance.

First, the natural occurrence of objects in the real-world leads to an imbalanced class distribution within each image. Re-sampling entire images only reinforces this imbalance, as each image retains its inherently unbalanced class distribution (Chang et al., 2021). Therefore, we introduce Rare Class Collage (RCC), which enhances the representation of underrepresented classes by creating a collage of cropped rare object images.

Second, while upsampling rare classes helps mitigate the imbalance, their infrequent occurrence in random training batches prevents the model from retaining the learned features. We propose Rare Class Focus (RCF), which ensures consistent exposure to rare classes by including at least one rare example in each training batch. Our method departs from the standard random sampling strategy, enhancing the ability of the model to handle challenging rare cases.

Third, labeled data is often implicitly assumed to be flawless, although ground truth label errors can range from 10% to 40% (Northcutt et al., 2021; Seedat et al., 2024), significantly hindering performance (Kuan & Mueller, 2022). Manual inspection to identify mislabeled data is time-consuming, error-prone, and frequently impractical due to the large dataset sizes (Tkachenko et al., 2023). We investigate the effect of noisy, false, and missing ground truth labels on performance by simulating two levels of errors. Our finding motivate the introduction of our Ground Truth Label Correction (GLC) method to refine noisy or incomplete ground truth labels by leveraging inference-time augmentation and teacher model predictions.

Finally, detections are usually filtered based on a score (Li et al., 2022b; Xu et al., 2021), with a *trade-off* between recall and precision (Zhang et al., 2022). For example, Xu et al. (2021) observe that setting a high threshold, such as 0.9 (Sohn et al., 2020), results in high precision. However, it also results in low recall, leading to many missed detections. In contrast, setting a lower threshold improves recall but retains many false detections. Although two-stage filtering, dynamic thresholding (Cai et al., 2021; Li et al., 2022c; Chen et al., 2023a), or using uncertainty as a score (Munir et al., 2021; Cai et al., 2021), can help reduce this trade-off, the resulting missing detections are still not addressed. Typically, all pseudo-labeled images are used to train the student model after the score-based filtering despite missing or false detections. This introduces noise and hinders effective learning (Xu et al., 2019; Yang et al., 2020). We therefore introduce Pseudo-Label Selection (PLS), a method that estimates the proportion of missing detections in an image using a novel metric. The metric also accounts for class distribution, ensuring that images with high missing detections are retained if they contain valuable learning signals from rare classes. This approach enables the filtering of low-quality pseudo-labeled images that could otherwise negatively impact model performance.

Our building blocks are designed to be as model- and framework-agnostic as possible, ensuring broad applicability to many SSOD frameworks and object detection tasks. Additionally, they add minimal computational overhead, ensuring feasibility for large-scale, real-world applications. Due to high labeling costs and abundant unlabeled data, SSL is crucial in real-world applications. Although some works (Cai et al., 2021; Han et al., 2021; Munir et al., 2021) explore the use of pseudo-labels for domain adaptation in autonomous driving, comprehensive analyses of the efficacy of pseudo-labeling remain limited. Our work addresses this gap by analyzing the key limitations of SSOD and proposing improvements that result in more effective and robust real-world SSOD. Our contributions are summarized as follows:

- We investigate the limitations of pseudo-labels in SSOD on real-world datasets, identifying factors hindering its effectiveness: (1) error propagation due to class imbalance, (2) noisy and erroneous ground truth labels, and (3) missing and false pseudo-labels.

- Based on the resulting insights, we develop novel and computationally efficient building blocks, each of which can be independently integrated into an SSOD framework to:

  - Balance the class distribution and mitigate error propagation on underrepresented classes.

- Assess the impact on performance by varying levels of noisy, false, and missing detections in ground truth labels and correcting them.
- Remove low-quality pseudo-labeled images post-filtering while considering class distribution.

## 2 Related Work

**Class Amplifiers:** Class imbalance often leads to rare classes being filtered out from pseudo-labels due to them receiving low confidence scores. The effectiveness of re-sampling, i.e., adjusting frequency (Yu et al., 2022; Chang et al., 2021), and re-weighting, i.e., adjusting sample weights in the loss function (Tantithamthavorn et al., 2018; Cui et al., 2019; Li et al., 2020a; Yu et al., 2022), strongly varies depending on detector type (Crasto, 2024). Re-weighting by inverse class frequency often leads to poor performance, particularly in highly imbalanced datasets where noisy rare samples receive disproportionately high weights (Phan & Yamamoto, 2020). On the other hand, synthetic re-sampling (Chawla et al., 2002; Chen et al., 2023b) and class-aware sampling (Shen et al., 2016) increase the frequency of rare classes artificially, reducing performance on common classes. Furthermore, entire image collages (Chen et al., 2020; Ly et al., 2023; Chen et al., 2023b) downscale images to increase object density, while copy-pasting approaches (Wang et al., 2018; Hong et al., 2019; Yan et al., 2022) can compromise spatial and contextual consistency. Ghiasi et al. (2021) demonstrate the effectiveness of copy-pasting for segmentation. However, object detection labels are rectangles that include background along with the object. This contrasts with instance segmentation, where objects can be seamlessly integrated into new images. Shen et al. (2016) ensure uniform class representation in the training batches for classification, but such a representation is challenging due to strongly imbalanced real-world data. Other class re-balancing techniques improve recall but may decrease precision due to overfitting on rare classes (Tantithamthavorn et al., 2018). To address these challenges, our Rare Class Collage (RCC) and Rare Class Focus (RCF) approaches increase the representation of rare classes while preserving spatial context. They ensure consistent and balanced exposure to rare classes during training without sacrificing performance on common classes.

**Label Filters:** Confirmation bias, where models overfit to incorrect labels, remains a critical challenge in SSOD (Arazo et al., 2020; Wang et al., 2021). Noisy and incomplete labels hinder effective learning. While several works attempt to mitigate pseudo-label noise using dynamic thresholding (Munir et al., 2021; Chen et al., 2022; Li et al., 2023; Chen et al., 2023a; Kimhi et al., 2024), consistency-based filtering (Li et al., 2022b; Yang et al., 2021; Yan et al., 2022), uncertainty-weighted filtering (Cai et al., 2021) or iterative refinement through multi-iteration predictions (Wang et al., 2018), these approaches often overlook noise in the ground truth labels, assuming perfectly labeled data. Seedat et al. (2024) use aleatoric uncertainty to identify noisy ground truth labels. Zhou et al. (2023) refine ground truth boxes by aggregating region proposals in two-stage detectors. These methods generally focus on box refinement and do not address missing or false labels. In contrast, inspired by consistency-based pseudo-label refinement (Li et al., 2022b) and consistency regularization (Li et al., 2022a), our Ground Truth Label Correction (GLC) mechanism simultaneously addresses noisy, missing, and false detections by evaluating ground truth labels through teacher predictions under inference-time augmentation.

As for pseudo-label quality, confidence-based filtering is typically used to retain high quality pseudo-labels (Liu et al., 2021), but high-confidence predictions do not always indicate accurate detections (Li et al., 2020b), and low-confidence scores may not always imply incorrect detections. The filtering process often leads to missing detections and noisy pseudo-labels remaining in the dataset, which amplifies errors over successive training iterations (Wang et al., 2021). The balance between the quality and quantity of pseudo-labels remains an unresolved challenge, as highlighted by conflicting works regarding the benefits of large volumes of pseudo-labeled data (Wang et al., 2021; Sohn et al., 2020). Wang et al. (2021) and Yang et al. (2021) argue that the volume of unlabeled data needs to be considered carefully, as more data does not necessarily lead to better performance. In contrast, Sohn et al. (2020) highlight the importance of large-scale unlabeled data in the context of SSL and do not observe a clear correlation between the accuracy of the pseudo-labels and the performance of their SSL approach. Our work investigates the impact of quality vs. quantity on performance. Furthermore, the impact of missing detections remains unclear. Xu et al. (2019) show that the performance of detectors declines significantly as the rate of missing detections increases. In contrast, Wu et al. (2018) and Chadwick & Newman (2019) argue for the robustness of detectors to it. We show that the impact of

missing detections on performance depends on the class of the missing objects. Building on this finding, we propose Pseudo-Label Selection (PLS). PLS proposes a metric that estimates the missing detection rate per image while accounting for class distribution, allowing for targeted selection of pseudo-labeled images post-filtering.

# 3 SSOD Building Blocks

In this section, we begin by outlining the student-teacher pseudo-labeling framework and then introduce our proposed building blocks. An overview of their integration into an SSOD framework is illustrated in Fig. 1, using the STAC framework (Sohn et al., 2020) as a representative example.

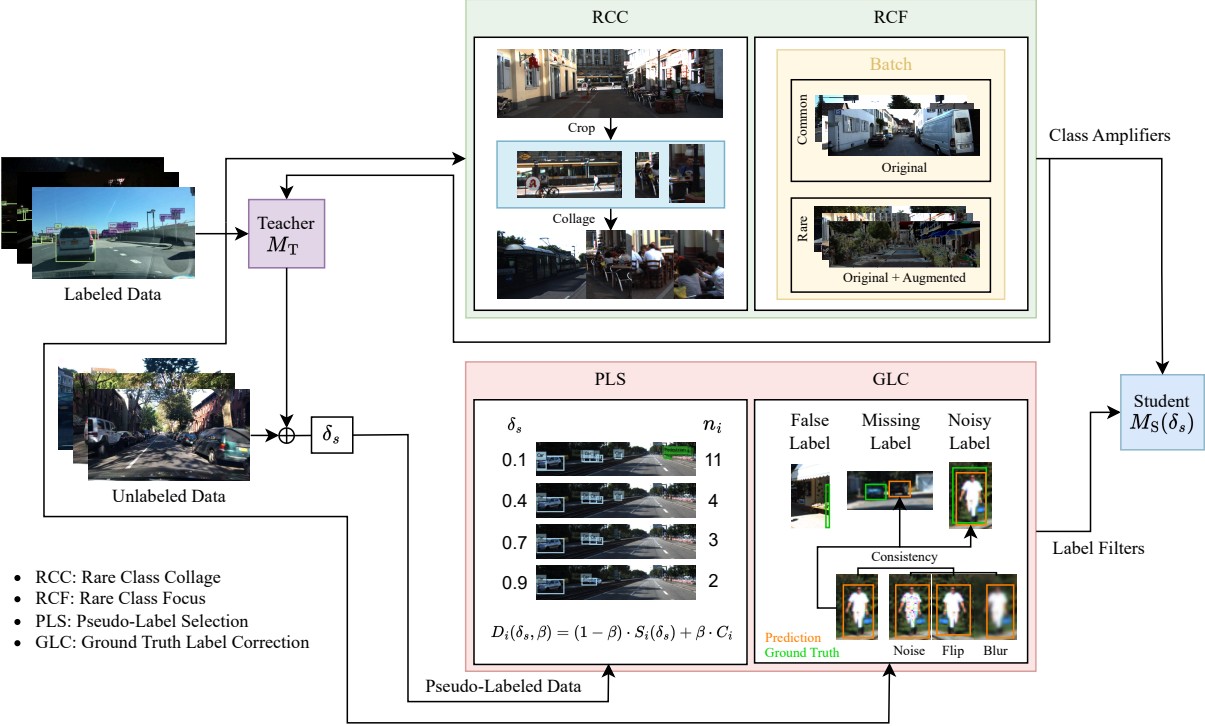

Figure 1: Our building blocks integrated into an exemplary SSOD framework. The teacher model $M_T$, trained on labeled data, generates pseudo-labels for unlabeled data, which are then filtered by a confidence threshold $\delta_s$. To address class imbalance, **Rare Class Collage (RCC)** (Section 3.2.1) crops instances of rare classes and combines them into collages, increasing their representation. **Rare Class Focus (RCF)** (Section 3.2.2) ensures each training batch contains common and rare classes, with augmented rare class images to boost their impact. **Ground Truth Label Correction (GLC)** (Section 3.3.1) corrects false, missing, and noisy labels by utilizing teacher prediction consistency across augmentations. **Pseudo-Label Selection (PLS)** (Section 3.3.2) removes pseudo-labeled images with many missing detections, estimated using our metric $D_i(\delta_s, \beta)$, which incorporates detection confidence and class rarity. Together, our methods enhance the ability of the student model $M_S(\delta_s)$ to learn effectively from both labeled and pseudo-labeled data, minimizing the propagation of errors from the teacher model.

## 3.1 Pseudo-Labeling

In SSOD, we are given a labeled dataset $\mathcal{D}_{\text{labeled}} = \{(x_i, b_i, c_i)\}_{i=1}^{m}$ of images $x_i$, bounding boxes $b_i$ and class labels $c_i$, for $i = 1, \ldots m$; and a larger unlabeled dataset $\mathcal{D}_{\text{unlabeled}} = \{x_i\}_{i=1}^{l}$, where $l \gg m$. The objective is to train a student model $M_S$ using both labeled data and pseudo-labels, with the latter generated by a

teacher model $M_{\text{T}}$ for the unlabeled data. Formally, the teacher model $M_{\text{T}}$, trained on $\mathcal{D}_{\text{labeled}}$, generates pseudo-labels for $\mathcal{D}_{\text{unlabeled}}$, resulting in a pseudo-labeled dataset $\mathcal{D}_{\text{pseudo}} = \{(x_i, \hat{b}_i, \hat{c}_i)\}_{i=1}^{l}$, where $\hat{b}_i$ and $\hat{c}_i$ represent the predicted bounding boxes and class labels. Following Sohn et al. (2020), pseudo-labeled images are augmented ($\mathcal{A}$), and the student model $M_{\text{S}}(\delta_s)$ is trained by minimizing the total loss given by

$$\ell_{\text{total}} = \frac{1}{m} \sum_{i=1}^{m} \ell_{\text{labeled}}(x_i, b_i, c_i) + \lambda \cdot \frac{1}{l} \sum_{i=1}^{l} \mathcal{I}(s \geq \delta_s) \ell_{\text{pseudo}}(\mathcal{A}(x_i), \hat{b}_i, \hat{c}_i), \tag{1}$$

where $\ell_{\text{labeled}}$ and $\ell_{\text{pseudo}}$ are the default loss terms of the detector calculated on $\mathcal{D}_{\text{labeled}}$ and $\mathcal{D}_{\text{pseudo}}$, respectively, and $\mathcal{I}$ is an indicator function that selects pseudo-labels based on their predicted confidence score $s$. Formally, $\mathcal{I}(s) = 1$ if the confidence score $s \geq \delta_s$ and zero otherwise. The hyperparameter $\lambda$, controlling the balance between the loss terms, is best set as $\lambda \in [1, 2]$ (Sohn et al., 2020).

## 3.2 Class Amplifiers

To alleviate class imbalance, we introduce two methods that boost the representation of rare classes in $\mathcal{D}_{\text{labeled}}$ and ensure their consistent inclusion during training for a balanced learning across all object classes.

### 3.2.1 Rare Class Collage (RCC)

RCC addresses class imbalance by increasing the representation of rare classes through collage generation. Rare objects identified in the ground truth labels are cropped and upscaled to fit the height of a new collage image, while preserving their original aspect ratios. These resized objects are arranged side by side on a blank canvas to create a collage primarily composed of rare class objects.

Let $\mathcal{C}_r$ denote the manually defined set of rare object classes. For each object of a class $\in \mathcal{C}_r$ in an image $x_i \in \mathcal{D}_{\text{labeled}}$, we crop an area around its bounding box $b_r = [b_r^x, b_r^y, b_r^w, b_r^h]$, where $b_r^x$ and $b_r^y$ are the top-left coordinates, and $b_r^w$ and $b_r^h$ represent the width and height. To determine the area, we sample a random scale factor $p_r$ from a uniform distribution $U(\gamma_{r,\min}, \gamma_{r,\max})$, resulting in an expanded bounding box $b_r' = [\max(0, b_r^x - p_r b_r^w), \max(0, b_r^y - p_r b_r^h), \min(w, b_r^x + (1 + p_r)b_r^w), \min(h, b_r^y + (1 + p_r)b_r^h)]$, where $w$ and $h$ are the width and height of the original image. This ensures that the cropping area based on the resized bounding box remains within the image boundaries.

After cropping, the rare objects are shuffled, resized, and sequentially pasted into a new collage image $x_i'$. The newly generated collages are added to $\mathcal{D}_{\text{labeled}}$, thereby increasing the representation of rare classes. Unlike previous works, RCC ensures improved generalization across specifically targeted rare classes without compromising performance on more common classes. It is also computationally efficient, as it processes only the images and their corresponding labels in $\mathcal{D}_{\text{labeled}}$.

### 3.2.2 Rare Class Focus (RCF)

Class imbalance in real-world data leads to a non-uniform distribution of classes across training batches, resulting in uneven gradient updates and limiting the ability of the detector to learn from rare class examples effectively. RCC increases the overall representation of rare classes, but it does not guarantee their balanced presence within each training batch. RCF further mitigates class imbalance by ensuring each training batch contains at least one example of a rare class. Unlike RCC, which explicitly defines rare classes, RCF stratifies $\mathcal{D}_{\text{labeled}}$ into common and rare examples based on class frequency, adjusting batch composition to include rare examples consistently. For that, it assigns a score $F_i$ to each image $x_i$ based on its class distribution. For an image $x_i$, containing $j$ classes, the score is computed as

$$F_i = \frac{1}{j} \sum_{k=1}^{j} \frac{1}{\log(f_k)}, \tag{2}$$

where $f_k$ is the frequency of class $k$ in $\mathcal{D}_{\text{labeled}}$. This inverse-logarithmic formulation ensures that rarer classes receive higher scores, emphasizing their importance in the learning process (Phan & Yamamoto, 2020). The

class scores are then linearly scaled within a predefined range $[1, \gamma_f]$, where $\gamma_f$ is the maximum score that controls the scaling strength.

To construct training batches, the score $F_i$ is computed for all images in $\mathcal{D}_{\text{labeled}}$. Images are ranked based on their scores, and the top $k$ images are classified as *rare*, while all others are considered as *common*. The value of $k$ is selected such that each batch contains at least one rare image, satisfying the condition $k \geq \frac{m}{B}$, where $m$ is the total number of images in $\mathcal{D}_{\text{labeled}}$ and $B$ is the batch size. Additionally, the rare set of images is augmented and shuffled twice to introduce diversity by adding a pair of rare samples to each batch. This ensures the batch size remains a multiple of two, ensuring efficient GPU utilization and stable batch normalization. Our RCF approach promotes balanced learning, providing adequate focus on rare classes, while preserving the ability of the model to generalize across all classes. Similar to RCC, the steps in RCF incur minimal computational overhead, as they operate solely on $\mathcal{D}_{\text{labeled}}$.

### 3.3 Label Filters

The quality of labels, both ground truth and pseudo-labels, is crucial to model performance. Leveraging the knowledge of the teacher model before training the student model helps prevent error propagation from label errors.

#### 3.3.1 Ground Truth Label Correction (GLC)

GLC is designed to refine noisy labels, remove false labels, and add missing ground truth (GT) labels by leveraging a trained teacher model and inference-time augmentations on the training set. This approach acknowledges that GT labels are not always reliable and aims to prevent erroneous GT labels from propagating through the training process to the student model. Specifically, the teacher model generates bounding box predictions on the original training images and their augmented versions post-training. They are then compared against GT labels to identify and correct errors before training the student.

Typical augmentations such as Gaussian blur, horizontal flipping, and Gaussian noise can be employed. The computational cost of GLC scales linearly with the number of augmentations, starting at a minimum of twice the base inference time. Corrections are determined based on the Intersection over Union (IoU) between predicted boxes $\hat{b}$ and GT boxes $b$. The core intuition is that consistent predictions from the teacher model across augmented versions of an image – despite the teacher not being trained on these augmentations – indicate the reliability of its detections. This consistency serves as an indicator of whether the detections may be used to correct GT errors. We define three error cases:

- **False GT:** A GT label $b$ that does not intersect with any prediction, even at a low confidence threshold $\delta_s = 0.1$, likely indicates a false label. In one-stage object detectors, anchors are defined densely across the grid; thus, any valid GT label should intersect with at least one anchor if there is a recognizable feature. Similarly, in two-stage detectors, the region proposal network generates region proposals around potential object locations, so any valid GT label should intersect with at least one proposal region if a relevant feature is present.

- **Missing GT:** If a predicted bounding box $\hat{b}$ appears consistently despite augmentations $(\mu_{\text{IoU}(\hat{b}, \mathcal{A}(\hat{b}))} > \gamma_c)$ but has no corresponding GT label, a GT label is likely missing for that object. Here, $\mu$ is the mean IoU between detections on the original and augmented images, and $\gamma_c$ is a predefined consistency threshold.

- **Noisy GT:** Typically, each GT box $b$ is paired with a predicted box $\hat{b}$ based on the highest IoU, provided the IoU exceeds the default threshold of 0.5. However, if the IoU between the predicted and GT box falls below an upper threshold $(\text{IoU}(\hat{b}, b) < \gamma_o)$, but the predicted box shows consistency despite augmentations $(\mu_{\text{IoU}(\hat{b}, \mathcal{A}(\hat{b}))} > \gamma_c)$, then the predicted box is considered more accurate and replaces the GT box.

For each of the three categories above, we simulate real-world labeling inaccuracies and evaluate the robustness of the model and the effectiveness of our GLC approach under synthetic error conditions as follows.

For false GT, random bounding boxes are added to simulate false detections. For each image, random boxes $\tilde{b}$ are generated, with width and height sampled from uniform distributions, $\tilde{b}^w \sim U(\gamma_{\tilde{b}^w,\min}, \gamma_{\tilde{b}^w,\max})$ and $\tilde{b}^h \sim U(\gamma_{\tilde{b}^h,\min}, \gamma_{\tilde{b}^h,\max})$, placed in non-overlapping positions relative to the existing GT boxes.

For missing GT, a percentage $\rho_{\mathrm{MGT}}$ of GT bounding boxes are randomly removed to simulate missing labels. The removed boxes are selected uniformly across $\mathcal{D}_{\mathrm{labeled}}$.

Finally, for noisy GT, bounding boxes are randomly perturbed in their dimensions and position. For a GT box $b = [b^x, b^y, b^w, b^h]$, the width and height are altered by $b^{w'} = b^w + \Delta b^w$ and $b^{h'} = b^h + \Delta b^h$, where $\Delta b^w$ and $\Delta b^h$ are sampled from $\{-\epsilon_b, \epsilon_b\}$. The center of the box is also shifted: $b^{x'} = b^x + \Delta b^x$ and $b^{y'} = b^y + \Delta b^y$, where $\Delta b^x = \pm\frac{\Delta w}{2}$ and $\Delta b^y = \pm\frac{\Delta h}{2}$.

### 3.3.2 Pseudo-Label Selection (PLS)

As outlined in Section 1, regardless of the choice or sophistication of the threshold $\delta_s$ which regulates the precision-recall trade-off in pseudo-labeling, pseudo-labeled images inherently contain missing detections. By estimating the missing detection rate (MDR) for each image, PLS identifies which pseudo-labeled images to remove after initial filtering with $\delta_s$.

We hypothesize that the distribution of confidence scores within an image is an indicator of missing detections. Specifically, the number of detections per image at different thresholds $\delta_s \in [0, 1]$ reflects the reliability of the detector on that image, correlating with the potential of missing detections. Given a confidence score threshold $\delta_s$, the number of detections at that threshold is denoted $n_i(\delta_s)$. We define the score metric for an image $x_i$ as

$$S_i(\delta_s) = \frac{n_i(\delta_s)}{n_i(\alpha)},$$

where $n_i(\alpha)$ represents the number of detections at a low reference threshold (e.g., $\alpha = 0.1$), providing a baseline for the total number of detections in an image. The value of $S_i$ increases as the potential of missing detections decreases.

To accommodate class imbalance, we recognize that pseudo-labeled images with a high MDR may still contain valuable learning signals if they include rare objects. To address this, we introduce a regularizing class metric $C_i$ which considers class rarity. To keep $C_i$ in the range $[0, 1]$, it is calculated as

$$C_i = \frac{1}{j} \sum_{k=1}^{j} \left(1 - \frac{f_k}{\sum_{i=1}^{m} n_i}\right),$$

where $j$ and $n_i$ are the number of classes and predictions in an image $x_i$, respectively, and $f_k$ is the frequency of class $k$ in $\mathcal{D}_{\mathrm{pseudo}}$. The value of $C_i$ increases as the rarity of the predicted objects increases.

The overall metric $D_i$ for an image $x_i$ proposed by our PLS method is then given by

$$D_i(\delta_s, \beta) = (1 - \beta) \cdot S_i(\delta_s) + \beta \cdot C_i, \tag{3}$$

where $\beta$ is a weighting factor balancing the contributions of missing detections and class rarity. The value of $D_i$ increases as the potential of missing detections decreases and class rarity increases. Therefore, images with low $D_i$ scores are filtered out to improve the quality of pseudo-labels used during student training. Eq. (3) offers a computationally efficient solution for pseudo-labeled images selection as it relies solely on the pseudo-labels.

## 4 Experiments

### 4.1 Implementation Details

To demonstrate the effectiveness of our data-centric methods, we select EfficientDet-D0 (Tan et al., 2020; Google, 2020) pre-trained on MS COCO (Lin et al., 2014) as our baseline detector. Experiments are conducted on two autonomous driving datasets: KITTI (Geiger et al., 2012) (7 classes, 20% random split for

validation) and BDD100K (Yu et al., 2020) (10 classes, 12.5% official split). Each training comprises 200 epochs with 8 batches and an input image resolution of 1024×512 pixels. All other hyperparameters of EfficientDet are set to their default values (Tan et al., 2020).

Our methods are also framework-agnostic. As an exemplary student-teacher framework, this work adopts STAC (Sohn et al., 2020) with $\lambda = 1$ in Eq. (1) due to its simplicity. For data augmentations ($\mathcal{A}$), we use RandAugment (Cubuk et al., 2020). Using AutoAugment (Cubuk et al., 2019) instead yields identical performance. We continue to denote the teacher model as $M_T$ and the student model as $M_S(\delta_s)$, which depends on the score threshold $\delta_s$. We observe a minimal inter-run variance and therefore average over a total of three training runs. We evaluate each proposed method individually to isolate its specific impact on SSOD performance.

## 4.2 SSOD Framework

**Impact of $\delta_s$ and Amount of Labeled Data:** We begin our experiments by examining key factors that influence the effectiveness of SSOD frameworks. These include the choice of confidence score threshold ($\delta_s$) and the ratio of labeled to unlabeled data. Fig. 2 shows a strong correlation between the choice of $\delta_s$ and the amount of labeled data used during training. As the quantity of labeled data increases, the pseudo-labels generated by the teacher model become more reliable, enabling the use of a higher $\delta_s$. While a lower threshold ($\delta_s = 0.4$) is optimal for setups with 15% labeled data in KITTI and 10% in BDD, a higher threshold ($\delta_s = 0.7$) outperforms the latter at 25% labeled data.

However, misconfiguring $\delta_s$, such as selecting 0.9 for 10% of BDD labeled, results in a high MDR since many pseudo-labels get filtered out. This leads to a significant performance drop of 3%, suggesting that both the amount of labeled data and the choice of $\delta_s$ must be carefully calibrated (also relative to each other) to achieve optimal performance. While SSL improves the mean Average Precision (mAP) on both datasets by up to a 2%, ineffective thresholding can lead to its degradation for $M_S$ by up to 3% compared to $M_T$. The latter is most pronounced when labeled data is significantly limited, and both labeled and pseudo-labeled data are noisy and unbalanced, as demonstrated by the consistent underperformance of $M_S$ at 1% of BDD labeled. This can be attributed to poor-quality pseudo-labels generated in such sparse settings, as shown in Table 1.

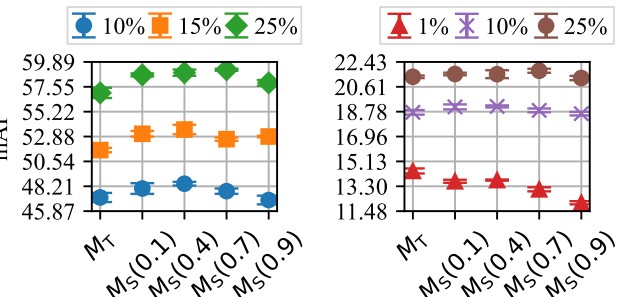

Figure 2: KITTI (left), BDD (right). Impact of the relationship between the confidence score threshold ($\delta_s$) and the proportion of labeled data on performance. A higher proportion of labeled data allows an effective increase in $\delta_s$. A misconfigured $\delta_s$ relative to the available labeled data results in a student ($M_S$) that underperforms its teacher ($M_T$).

To avoid misconfiguring $\delta_s$, we generally recommend based on our experiments to calculate the average score across matched detections with GT labels in the validation set to effectively select the minimum $\delta_s$ for pseudo-labels. We select $\delta_s = 0.4$ as the default value since it aligns with the default setting of EfficientDet (Tan et al., 2020) and matches the average score ($0.36 \pm 0.1$) across correctly matched detections on the validation sets of both KITTI and BDD.

**Impact of Pseudo-Label Quality:** We continue our experiments on the SSOD framework by investigating the quality of pseudo-labels and its impact on performance. We identify three primary factors that affect it: noisy ($N_D$), false ($F_D$), and missing detections ($M_D$). As shown in Table 1, $M_T$ achieves an mAP of 47.16% on KITTI and 14.43% on BDD. The default $M_S(0.4)$ shows an increase of 1.29% on KITTI but a decrease of 0.97% on BDD. This poor performance on BDD is a result of poorer pseudo-label quality, as demonstrated by metrics such as classification accuracy (mACC), mean Intersection over Union (mIoU), MDR, and uncategorized detection rate (UDR). MDR represents instances that are not detected, while UDR refers to detections that may be correct but lack a corresponding GT label or are entirely false.

Table 1: KITTI (top, 10% labeled), BDD (bottom, 1% labeled). Impact of pseudo-label quality on performance, with $N_D$s, $F_D$s, and $M_D$s removed ($\times$) from the pseudo-labels. A dash (-) indicates non-applicability.

| | $N_D$ | $F_D$ | $M_D$ | mAP (%) | MDR (%) | UDR (%) | mACC (%) | mIoU (%) | $n_i$ | $m+l$ |
|---|---|---|---|---|---|---|---|---|---|---|
| $M_T$ | - | - | - | 47.16 | - | - | - | - | 3069 | 598 |
| $M_S(0.4)$ | - | - | - | 48.45 | 23.26 | 7.10 | 97.24 | 85.92 | 23453 | 5866 |
| $M_S(0.4)$ | $\times$ | - | - | **52.21** | 29.54 | 6.66 | 99.30 | 95.97 | 23453 | 5866 |
| $M_S(0.4)$ | - | $\times$ | - | 48.95 | 23.26 | 0.00 | 97.24 | 85.92 | 21789 | 5861 |
| $M_S(0.4)$ | - | - | $\times$ | 47.47 | 00.00 | 4.92 | 97.83 | 87.40 | 6540 | 2668 |
| $M_T$ | - | - | - | 14.43 | - | - | - | - | 12365 | 698 |
| $M_S(0.4)$ | - | - | - | 13.46 | 64.03 | 14.63 | 96.36 | 81.15 | 528818 | 69021 |
| $M_S(0.4)$ | $\times$ | - | - | 13.59 | 64.03 | 14.63 | 98.83 | 90.30 | 528818 | 69021 |
| $M_S(0.4)$ | - | $\times$ | - | **13.87** | 64.03 | 00.00 | 96.36 | 81.23 | 450202 | 69021 |

The quality of pseudo-labels has a more significant impact on student model performance than their quantity. Despite the number of detections per image ($n_i$) being approximately 23 times higher in BDD compared to KITTI, the quality of pseudo-labels remains crucial. The 12 times larger number of unlabeled images ($l$) in BDD at a comparable number of labeled images ($m$, 598 for KITTI vs. 698 for BDD) could not compensate for lower pseudo-label quality.

Further analysis shows that the influence of $N_D$, $F_D$, and $M_D$ in pseudo-labeled data is closely tied to the quality of the GT labels. On KITTI, which features higher-quality images and labels, removing $N_D$s leads to an approximate 4% increase in mAP. However, excluding images with $M_D$s lowers performance by reducing the total number of images $l$ in $\mathcal{D}_{\text{unlabeled}}$ and detections $n_i$. This reduction compromises the reliability of $\ell_{\text{pseudo}}$ and disrupts the balance between $\ell_{\text{labeled}}$ and $\ell_{\text{pseudo}}$. Since both losses are weighted equally in the total loss (Eq. (1)), the negative impact of the low quality of the pseudo-labels is further amplified by their limited quantity. Pseudo-label quality generally has a more significant impact than quantity, but quantity becomes crucial when GT label quality is high. Furthermore, some pseudo-labeled images remain valuable even if they contain $M_D$s, particularly when they contain rare objects as demonstrated in Section 4.4. Meanwhile, removing $F_D$s from BDD boosts performance more than removing $N_D$s, underscoring the challenges of noisy datasets. In BDD, no images exist without $M_D$s, thereby preventing an analysis of the impact of their complete absence.

These observations emphasize the need to maintain a careful balance between noise, precision, and recall in pseudo-labels based on the characteristics of $\mathcal{D}_{\text{labeled}}$. We also test advanced pseudo-label selection strategies, such as uncertainty-based thresholding with class-specific weighting, but observe no direct gains in mAP over using $\delta_s$ for filtering. This is primarily due to the rarity of some classes in the labeled set, which adversely impacts the performance of $M_T$ on them, hence propagating errors to $M_S$. As a result, for SSL to be effective in object detection, a class-specific approach is necessary, motivating the development of our Rare Class Collage (RCC) and Rare Class Focus (RCF) methods.

*Summary: our experiments reveal that optimal SSOD performance depends on careful calibration of the confidence threshold $\delta_s$ with the amount of labeled data. Furthermore, pseudo-label quality is generally more influential than quantity, unless $\mathcal{D}_{labeled}$ is of high quality and $M_T$ is well-trained. KITTI benefits from noise reduction in pseudo-labels, while BDD sees improvement with reduced false detections, underscoring the importance of dataset-specific adjustments.*

### 4.3 Class Amplifiers

Due to limited labeled data in SSL, class imbalance significantly impacts model performance. In addition to our proposed RCC and RCF methods, we evaluate two baselines that address this: classical image-level re-sampling and re-weighting, as detailed in Section 2.

**Imbalance Strength:** Class imbalance is evident in Fig. 3, showing the frequency $f_k$ of classes in KITTI and BDD. The imbalance spans a 260-fold difference between the most and least frequent classes. In RCC,

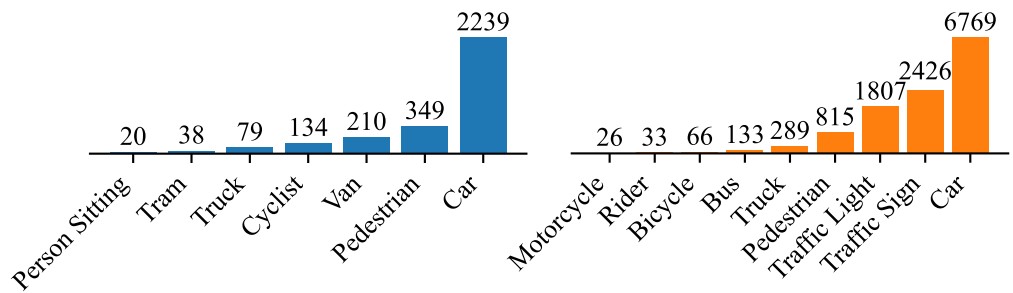

Figure 3: KITTI (left, 10% labeled), BDD (right, 1% labeled). Class frequency $f_k$ for each class in $\mathcal{D}_{\text{labeled}}$.

we target the rare classes "person sitting" and "tram" for KITTI, and "rider", "motorcycle", and "bicycle" for BDD to boost their representation. An example collage is visualized in Fig. 4 for each dataset. Unlike RCC, RCF does not explicitly target specific classes but instead stratifies $\mathcal{D}_{\text{labeled}}$ by using a class score that accounts for the frequency of all classes, as defined in Eq. (2).

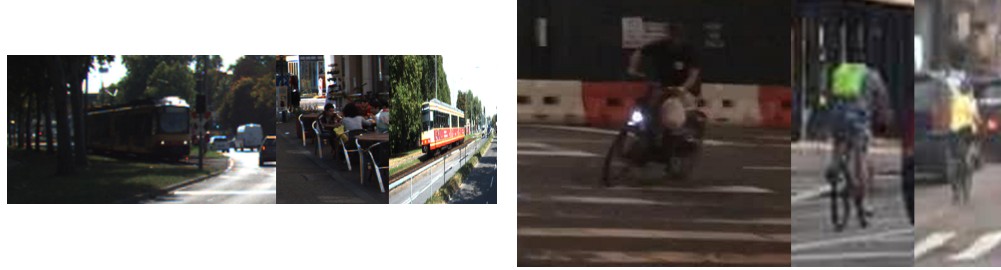

Figure 4: KITTI (left), BDD (right). Example collages with scaling factors $\gamma_{r,\text{min}} = 0.25$ and $\gamma_{r,\text{max}} = 0.75$.

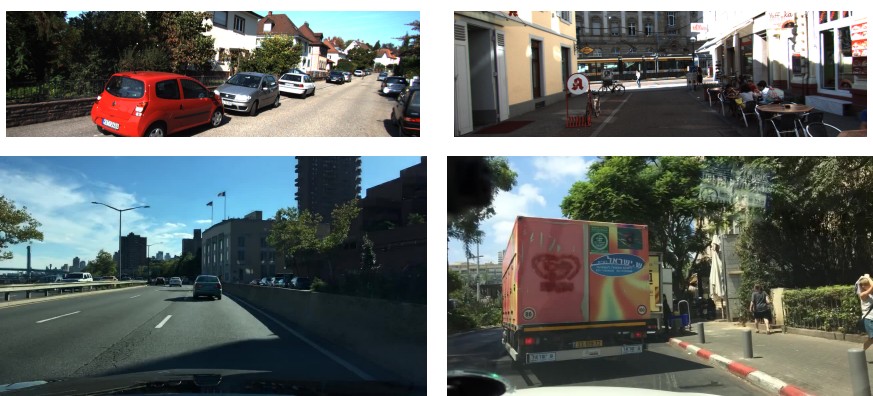

Figure 5: KITTI (top), BDD (bottom). Batch structuring via RCF with images categorized as common (left) predominantly containing cars and images categorized as rare (right) including the rare classes "tram", "truck", "pedestrian" and "person_sitting".

**Overall Performance of RCC and RCF:** Table 2 highlights the importance of a per-class perspective when analyzing SSOD performance. While both RCC and RCF improve $M_{\text{T}}$ performance, only RCF consistently benefits $M_{\text{S}}$. RCC targets specific classes, resulting in a limited number of collage images (18 for KITTI and 43 for BDD) based on our choice of classes. The impact of the additional images becomes negligible during the training of $M_{\text{S}}$ due to the thousands of additional pseudo-labeled images. Furthermore, the class-specific improvements from RCC have a less significant impact on mAP, as mAP averages performance across all classes by default.

Nevertheless, the impact of class balancing through our methods is clear: using RCC for targeted collaging and RCF for structured batch training, the mAP of $M_{\text{T}}$ is improved by up to 1% from each approach with no additional manual labeling or changes to the core training process. An example of the batch structuring via RCF is visualized in Fig. 5. Fig. 6 demonstrates that RCC effectively increases the frequency of targeted rare classes by approximately 300% while minimally affecting the distribution of common classes (up to 3% increase). In contrast, re-sampling introduces a greater bias toward common classes (up to 10% increase) while achieving a comparatively smaller boost in the representation of rare classes (only 200%).

Table 2: KITTI (left, 10% labeled), BDD (right, 1% labeled). Impact of RCF and RCC on mAP.

|  | RCF | RCC | mAP | |
|---|---|---|---|---|
| $M_{\text{T}}$ | - | - | $47.16 \pm 0.43$ | $14.43 \pm 0.18$ |
| $M_{\text{T}}$ | ✓ | - | $\mathbf{48.11 \pm 0.15}$ | $14.65 \pm 0.16$ |
| $M_{\text{T}}$ | - | ✓ | $47.49 \pm 0.29$ | $\mathbf{15.00 \pm 0.06}$ |
| $M_{\text{S}}(0.4)$ | - | - | $48.45 \pm 0.18$ | $13.76 \pm 0.04$ |
| $M_{\text{S}}(0.4)$ | ✓ | - | $\mathbf{49.56 \pm 0.13}$ | $\mathbf{14.27 \pm 0.05}$ |
| $M_{\text{S}}(0.4)$ | - | ✓ | $48.41 \pm 0.11$ | $14.26 \pm 0.15$ |

**Per-Class Performance of RCC and RCF:** We investigate both methods from a per-class perspective. Table 3 compares the performance of RCC and RCF against re-weighting (RW) and re-sampling entire images (RS). For RW, we leverage $F_i$ in Eq. (2) to weight images in $\mathcal{D}_{\text{labeled}}$ as described in Section 2. We use the mean weight per image since alternative approaches, such as per-detection weighting and using sum or max values per image, were less effective. The higher mAP of RCF observed in Table 2 is further supported by consistent performance gains across individual classes. However, while RCF requires changes to the data loader, RCC remains a more straightforward, data-centric approach that consistently outperforms RS and RW on targeted rare classes.

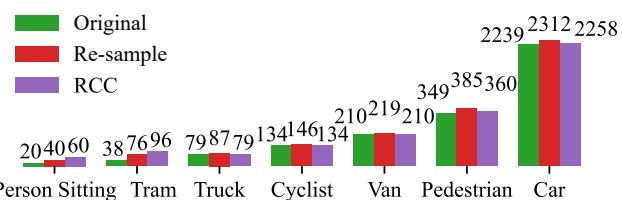

Figure 6: KITTI. Class frequency $f_k$ after applying RCC vs. re-sampling of entire images. RCC increases $f_k$ for targeted classes ("person sitting" and "tram") without over-representing common classes.

The impact on other classes varies across datasets. On KITTI, RCC reduces the AP on common classes by up to 1%, while on BDD it consistently improves it by up to 1% across all classes (see Table 3). This variation can be attributed to the strong model performance on KITTI, where additional collage-based augmentation may introduce unnecessary variability. This is also evident in the total sum ($\sum$) across all classes, where RCC achieves an additional 1% improvement in total AP on BDD compared to KITTI. On the other hand, RCF enhances performance across all classes as it maintains the original object resolution. RW performs best on KITTI (total increase of 10.53% in AP across all classes) but performs worse on BDD (decrease of 4.49%), particularly struggling with common classes. This is due to its disregard for label noise when weighting samples, and BDD containing higher noise levels compared to KITTI. Challenging samples are assigned high weights, hindering learning on easier and more common samples. Meanwhile, RS marginally improves the AP on rare classes by up to 1% at the cost of degrading it on common classes by up to 1%. This decline occurs because RS introduces challenging images that hinder the effective optimization of Eq. (1). It is worth mentioning that we test re-sampling both with and without augmenting the images using RandAugment (Cubuk et al., 2020). Given the consistent underperformance of RS compared to other methods in Table 3, we exclude it from the analysis in Table 4. Furthermore, the "train" class is excluded from the analysis on BDD because there is only one instance of a train in the labeled set at 1% and it is occluded, resulting in an AP of 0% for that class irrespective of the class-balancing approach used.

Both tables (Table 3 for $M_{\text{T}}$ and Table 4 for $M_{\text{S}}(0.4)$) show similar trends with RCF yielding the highest improvements for rare classes by up to 5% AP without significantly reducing performance on common classes (less than 1% AP). This indicates that addressing class imbalance in $\mathcal{D}_{\text{labeled}}$ is beneficial, regardless of whether the training includes $\mathcal{D}_{\text{pseudo}}$. However, the impact is diminished for $M_{\text{S}}(0.4)$ (9.15% vs 3.16% total increase in AP across all classes), as the contribution of $\ell_{\text{labeled}}$ to the overall training is reduced. The limited improvement with RCC on the AP of the student (up to 2% vs. 5% for the teacher) can be

Table 3: KITTI (top, 10% labeled), BDD (bottom, 1% labeled). Impact of RS, RW, RCF ($\gamma_f = 20$), and RCC ($\gamma_{r,\min} = 0.25$, $\gamma_{r,\max} = 0.75$) on per-class AP of $M_\mathrm{T}$ and the total sum across all classes ($\sum$). Green arrows (↑) indicate a performance increase, while red arrows (↓) denote a decrease compared to the mean AP of the original $M_\mathrm{T}$ across the multiple runs (shown in the first row of each dataset). The values adjacent to the arrows are the absolute differences to the mean APs. Colored columns highlight the classes targeted by RS and RCC. Classes are ordered left to right from rarest to most common.

| RS | RW | RCF | RCC | Person Sitting | Tram | Truck | Cyclist | Van | Pedestrian | Car | - | - | $\sum$ |
|---|---|---|---|---|---|---|---|---|---|---|---|---|---|
| - | - | - | - | 22.89 | 43.64 | 60.14 | 45.03 | 51.70 | 36.80 | 65.99 | - | - | |
| ✓ | - | - | - | ↑1.30 | ↑0.95 | ↓0.88 | ↓0.42 | ↓0.00 | ↓0.52 | ↓0.10 | - | - | ↑0.33 |
| - | ✓ | - | - | ↑2.72 | ↑2.64 | ↓1.08 | ↑0.26 | ↑0.76 | ↑0.32 | ↑0.31 | - | - | ↑10.53 |
| - | - | ✓ | - | ↑3.18 | ↑4.32 | ↑1.03 | ↑0.17 | ↓0.25 | ↑0.81 | ↓0.11 | - | - | ↑9.15 |
| - | - | - | ✓ | ↑4.92 | ↑1.59 | ↓1.05 | ↓0.44 | ↓0.99 | ↓0.23 | ↓0.30 | - | - | ↑3.50 |
| | | | | Motor-cycle | Rider | Bicycle | Bus | Truck | Pedestrian | Traffic Light | Traffic Sign | Car | |
| - | - | - | - | 4.59 | 6.66 | 9.59 | 24.72 | 23.02 | 17.07 | 7.83 | 14.71 | 35.62 | |
| ✓ | - | - | - | ↑0.96 | ↑2.02 | ↓0.81 | ↓0.05 | ↑0.41 | ↓0.04 | ↓0.00 | ↑0.21 | ↓0.21 | ↑2.49 |
| - | ✓ | - | - | ↓0.78 | ↑0.91 | ↓1.27 | ↓2.18 | ↓0.54 | ↓0.88 | ↑0.06 | ↑0.28 | ↓0.09 | ↓4.49 |
| - | - | ✓ | - | ↑1.00 | ↑1.60 | ↓0.05 | ↓0.45 | ↑0.52 | ↑0.26 | ↑0.19 | ↑0.29 | ↓0.20 | ↑3.16 |
| - | - | - | ✓ | ↑1.90 | ↑1.88 | ↓0.29 | ↑0.71 | ↑0.24 | ↑0.01 | ↑0.09 | ↑0.02 | ↓0.02 | ↑4.54 |

attributed to the small number (18 for KITTI) of generated collage images relative to the large pool of added pseudo-labeled images (5866 for KITTI). Nonetheless, RCC still improves performance on targeted rare classes. Meanwhile, RW significantly reduces performance, particularly for rare classes, with drops of up to 3% in AP. Our findings support our hypothesis regarding error propagation: $M_\mathrm{S}(0.4)$ underperforms $M_\mathrm{T}$ on certain inadequately-learned rare classes such as "motorcycle" in BDD, yet improves performance on well-learned classes such as "car" and "traffic light". Both RCF and RCC effectively mitigate this issue, consistently improving performance on rare classes across both datasets while minimizing the impact on common classes.

Table 4: KITTI (top, 10% labeled), BDD (bottom, 1% labeled). Impact of RW, RCF ($\gamma_f = 20$), and RCC ($\gamma_{r,\min} = 0.25$, $\gamma_{r,\max} = 0.75$) on per-class AP of $M_\mathrm{S}(0.4)$.

| RW | RCF | RCC | Person Sitting | Tram | Truck | Cyclist | Van | Pedestrian | Car | - | - | $\sum$ |
|---|---|---|---|---|---|---|---|---|---|---|---|---|---|
| - | - | - | 25.05 | 47.60 | 61.21 | 47.36 | 51.60 | 37.45 | 67.60 | - | - | |
| ✓ | - | - | ↓2.50 | ↑2.54 | ↓0.69 | ↓1.21 | ↑0.52 | ↓0.93 | ↓1.45 | - | - | ↓3.72 |
| - | ✓ | - | ↑3.26 | ↑5.03 | ↑2.13 | ↓0.34 | ↓0.01 | ↑1.31 | ↓0.09 | - | - | ↑11.29 |
| - | - | ✓ | ↑1.05 | ↓0.05 | ↑1.21 | ↓0.42 | ↓0.62 | ↑0.37 | ↓0.34 | - | - | ↑1.20 |
| | | | Motor-cycle | Rider | Bicycle | Bus | Truck | Pedestrian | Traffic Light | Traffic Sign | Car | |
| - | - | - | 4.13 | 5.62 | 7.48 | 23.89 | 22.78 | 13.63 | 9.14 | 15.37 | 36.55 | |
| ✓ | - | - | ↓0.75 | ↓0.32 | ↓1.64 | ↓2.40 | ↓1.90 | ↓0.89 | ↓0.22 | ↓0.20 | ↓0.56 | ↓8.88 |
| - | ✓ | - | ↑1.43 | ↑0.48 | ↑0.73 | ↓0.02 | ↑0.87 | ↑2.17 | ↓0.62 | ↑0.02 | ↓0.24 | ↑4.82 |
| - | - | ✓ | ↑0.91 | ↑1.24 | ↑0.40 | ↓0.25 | ↑0.36 | ↑0.71 | ↓0.10 | ↑0.25 | ↑0.07 | ↑3.59 |

*Summary: class imbalance strongly impacts SSOD performance, particularly with limited labeled data, and is effectively addressed by our RCC and RCF methods. RCC targets specific rare classes in KITTI and BDD, while RCF achieves balanced representation across all classes during training. Therefore, if performance on common classes is less critical, RCC is more beneficial. However, if maintaining performance on common classes is a priority, RCF is the better choice. Both methods consistently outperform traditional re-sampling and re-weighting, underscoring the critical role of per-class analysis when evaluating SSOD.*

### 4.4 Label Filters

First, we investigate the impact of the quality of labeled data on performance and the effectiveness of our GLC method. Second, we evaluate our proposed metric $D_i$ and the advantages of selecting pseudo-labeled images with our PLS method post-filtering via $\delta_s$.

**GLC.** We first evaluate the impact of missing ($\mathrm{M_{GT}}$), false ($\mathrm{F_{GT}}$) and noisy ($\mathrm{N_{GT}}$) GT labels on the performance of $M_\mathrm{T}$. Table 5 presents the number of discovered errors in KITTI and BDD using GLC. Given their limited number in both datasets (up to 3% of all detections), GLC only slightly improves the mAP by around 0.5%. Examples of detected GT errors and their correction are visualized in Fig. 7 for both datasets.

Table 5: KITTI (left, 10% and 15% labeled), BDD (right, 1% and 10% labeled). Total number of detections $\sum_{i=1}^{m} n_i$ in $\mathcal{D}_{\mathrm{labeled}}$ and identified GT errors.

| % | $\sum_{i=1}^{m} n_i$ | $\mathrm{F_{GT}}$ | $\mathrm{M_{GT}}$ | % | $\sum_{i=1}^{m} n_i$ | $\mathrm{F_{GT}}$ | $\mathrm{M_{GT}}$ |
|---|---|---|---|---|---|---|---|
| 10 | 3069 | 14 | 20 | 1 | 4661 | 20 | 20 |
| 15 | 12365 | 380 | 123 | 10 | 126726 | 2772 | 1608 |

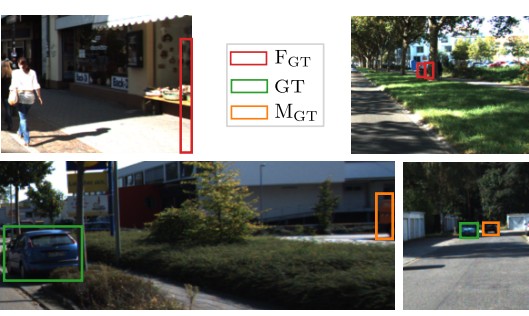 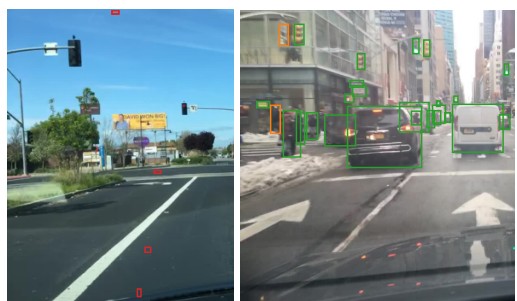

Figure 7: KITTI (left), BDD (right). Examples of GT label errors identified and corrected via our GLC approach. Both human annotators and model-assisted labeling tools often fail to label distant or occluded objects. Moreover, they frequently make errors, such as drawing tiny, incorrect bounding boxes scattered across the images or misplacing boxes near the image edges.

**Impact of GT Quality:** To further understand the impact of GT quality on model performance, we introduce two levels of synthetic GT errors as depicted in Fig. 8. This allows for a controlled analysis of label errors, highlighting the sensitivity of the model to the quality of the labeled data and emphasizing the advantages of our proposed GLC method. The two levels include:

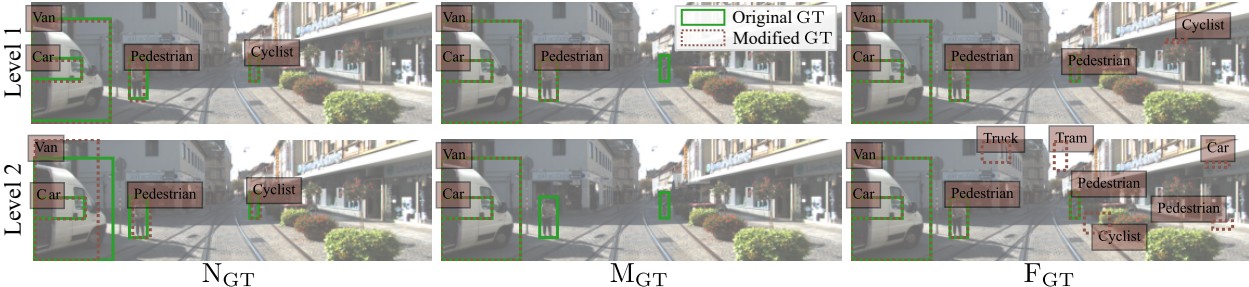

Figure 8: KITTI. Visualization of two levels of synthetic GT errors: Level 1 (+) and Level 2 (++).

- Level 1 (+): $\rho_{\mathrm{M_{GT}}} = 20\%$ dropped GT boxes, added one mistake per image ($\gamma_{\tilde{b}^w,\min} = 10$ pixels and $\gamma_{\tilde{b}^w,\max} = 100$ pixels), and noise applied to 20% of images with a perturbation factor $\epsilon_b = 0.1$.

- Level 2 (++): $\rho_{\mathrm{M_{GT}}} = 50\%$ dropped GT boxes, added five mistakes per image ($\gamma_{\tilde{b}^w,\min} = 10$ pixels and $\gamma_{\tilde{b}^w,\max} = 100$ pixels), and noise applied to 20% of images with a perturbation factor $\epsilon_b = 0.2$.

Table 6: KITTI (top, left 10%, right 15% labeled), BDD (bottom, left 1%, right 10% labeled). Impact of synthetic errors on mAP of $M_{\text{T}}$ without ($\times$) and with GLC ($\checkmark$). The errors include noisy ($N_{\text{GT}}$), false ($F_{\text{GT}}$), and missing ($M_{\text{GT}}$) GT labels, evaluated at two intensity levels: Level 1 (+) and Level 2 (++).

| $N_{\text{GT}}$ | $F_{\text{GT}}$ | $M_{\text{GT}}$ | mAP | | | |
|---|---|---|---|---|---|---|
| | | | $\times$ | $\checkmark$ | $\times$ | $\checkmark$ |
| - | - | - | $47.16 \pm 0.43$ | | $51.61 \pm 0.21$ | |
| + | - | - | $39.38 \pm 0.21$ | $45.65 \pm 0.32$ | $45.86 \pm 0.13$ | $49.81 \pm 0.20$ |
| ++ | - | - | $34.99 \pm 0.73$ | $43.08 \pm 0.37$ | $37.62 \pm 0.40$ | $46.77 \pm 0.89$ |
| - | + | - | $41.64 \pm 0.44$ | $46.76 \pm 0.58$ | $47.28 \pm 0.96$ | $51.54 \pm 0.56$ |
| - | ++ | - | $33.32 \pm 0.77$ | $46.56 \pm 0.33$ | $37.54 \pm 0.48$ | $51.21 \pm 0.26$ |
| - | - | + | $41.91 \pm 1.02$ | $43.88 \pm 0.02$ | $46.19 \pm 0.42$ | $48.63 \pm 0.15$ |
| - | - | ++ | $29.43 \pm 0.18$ | $39.88 \pm 0.83$ | $32.75 \pm 0.93$ | $42.89 \pm 0.39$ |
| - | - | - | $14.43 \pm 0.18$ | | $18.73 \pm 0.17$ | |
| + | - | - | $11.78 \pm 0.61$ | $13.57 \pm 0.01$ | $15.85 \pm 0.02$ | $17.05 \pm 0.83$ |
| ++ | - | - | $11.32 \pm 0.18$ | $13.54 \pm 0.02$ | $15.06 \pm 0.34$ | $16.99 \pm 0.71$ |
| - | + | - | $14.17 \pm 0.07$ | $14.12 \pm 0.24$ | $18.25 \pm 0.01$ | $18.88 \pm 0.04$ |
| - | ++ | - | $13.02 \pm 0.08$ | $14.17 \pm 0.02$ | $17.25 \pm 0.13$ | $18.81 \pm 0.17$ |
| - | - | + | $12.68 \pm 0.22$ | $13.39 \pm 0.21$ | $16.65 \pm 0.20$ | $17.21 \pm 0.11$ |
| - | - | ++ | $09.50 \pm 0.23$ | $12.02 \pm 0.03$ | $13.93 \pm 0.35$ | $15.54 \pm 0.02$ |

The performance impact of the selected error levels on $M_{\text{T}}$ is presented in Table 6. All error types substantially reduce the mAP, with a reduction of up to 18%. $M_{\text{GT}}$s have the most detrimental effect across all tested data regimes and datasets. As for $F_{\text{GT}}$s, even introducing a single error per image results in up to a 6% drop in mAP, demonstrating the high sensitivity of the model to GT errors. This model vulnerability remains consistent irrespective of the data regime or dataset characteristics. For instance, despite KITTI having four times fewer detections (3069 in 598 images) compared to BDD (12365 in 698 images, see Table 1), the performance reduction is comparable at both levels. The mAP drops relatively to its original value by 17% and 38% on KITTI and 18% and 34% on BDD at Levels 1 and 2, respectively.

**Effectiveness of GLC:** GLC mitigates the impact of GT label errors by correcting them before training $M_{\text{S}}$ or for retraining $M_{\text{T}}$. Correcting $F_{\text{GT}}$s is found to be more straightforward than recovering $M_{\text{GT}}$s due to overlapping GT labels, as GLC assumes for the latter no overlap between consistent detections and GT labels. As for $N_{\text{GT}}$s, GLC restores significant performance, raising mIoU from 83.98% to 90.97% and mAP from 39.38% to 45.65% on KITTI (10% labeled) after correction at Level 1.

**PLS.** Given the importance of addressing $M_{\text{GT}}$s and the abundance of $M_{\text{D}}$s in $\mathcal{D}_{\text{pseudo}}$ of real-world datasets such as BDD, we introduce PLS. The mIoU of $M_{\text{T}}$ trained on 10% of KITTI is around 88% for common classes such as "car", but drops to 76% for rare classes such as "person sitting". Similarly, the mACC is 100% for common classes vs. 96% for rare ones. This disparity motivates the consideration of the class distribution and the estimated MDR when selecting pseudo-labeled images with PLS.

**Evaluating $D_i$:** Fig. 9 demonstrates the effectiveness of our proposed metric $S_i$ in identifying images with high $M_{\text{D}}$s, achieving an AUC of 90% on KITTI and 91% on BDD, significantly outperforming the average score per image ($\mu_s$) at 74%, and the number of detections per image ($n_i$) at 66%.

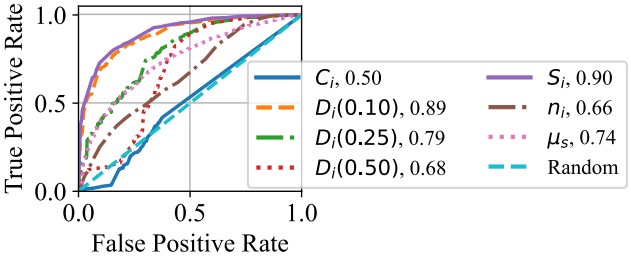

Figure 9: KITTI. ROC curve for identifying images with more than 50% $M_{\text{D}}$s ($\delta_s = 0.9$). Our metrics $S_i$ and $D_i$ outperform baseline metrics such as the average score per image ($\mu_s$) and the number of detections per image ($n_i$) (see AUC values).

As expected, class distribution does not correlate with the MDR (50% AUC). Still, it is beneficial when incorporated into our metric $D_i$, as it allows for the inclusion of images containing rare classes without compromising effectiveness in recognizing $M_D$s. This is evidenced by $D_i(0.10)$ and $D_i(0.25)$ still outperforming $\mu_s$ and $n_i$ despite the increase of $\beta$. The results are consistent across both KITTI and BDD, underscoring the robustness of our approach.

**Effectiveness of PLS:** Fig. 10 qualitatively compares the lowest and highest scoring images based on $S_i$. The contrast between these examples provides insights into the effectiveness of PLS in discriminating between high-quality and low-quality pseudo-labeled images, reinforcing its reliability in pseudo-label selection.

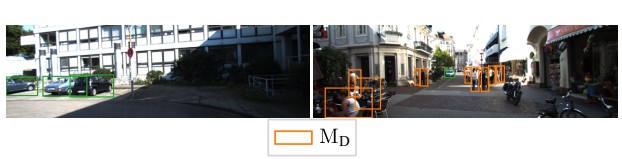 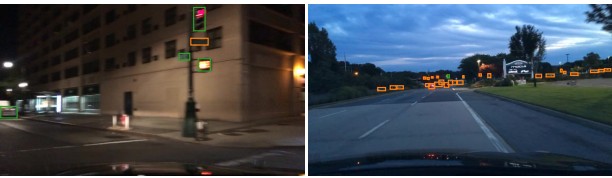

Figure 10: KITTI (left), BDD (right). For each dataset, the left image represents the highest $S_i$, while the right image represents the lowest $S_i$ (poor detection quality). Predictions (in green) are filtered at $\delta_s = 0.9$.

Fig. 11 summarizes the PLS results. At 1% of BDD labeled and $\delta_s = 0.4$, we compare mAP, MDR, and UDR across different configurations ($\beta = 0.1$ and $0.2$). We demonstrate that PLS consistently outperforms random selection in mAP and MDR/UDR, confirming that PLS effectively identifies high-quality pseudo-labeled images. PLS improves the performance of $M_S(0.4)$ to match or even surpass the performance of $M_T$ (see Table 1) by up to 0.5% mAP. Our method effectively filters out images with high MDR and UDR (around 80% and 16% in the removed set, compared to 60% and 14.5% in the remaining pseudo-labeled images), indicating a correlation between our metric and both the MDR and UDR.

Moreover, the model trained on the remaining 50% of $\mathcal{D}_{\text{pseudo}}$ post-selection using our $D_i$ metric outperforms the original student trained on the full $\mathcal{D}_{\text{pseudo}}$ and the students trained on a random 50% subset or the removed 50%. Increasing $\beta$ removes fewer rare classes, further underscoring the importance of a class-balanced perspective in SSL.

*Summary: GT and pseudo-label quality significantly impacts model performance. Synthetic errors reveal high model sensitivity to GT label quality, with even minor errors causing notable performance declines. Our GLC method effectively identifies and corrects GT label errors, while PLS filters low-quality pseudo-labeled images. By preventing error propagation and leveraging the learned knowledge of $M_T$, GLC and PLS improve the effectiveness of SSOD and therefore $M_S$ performance.*

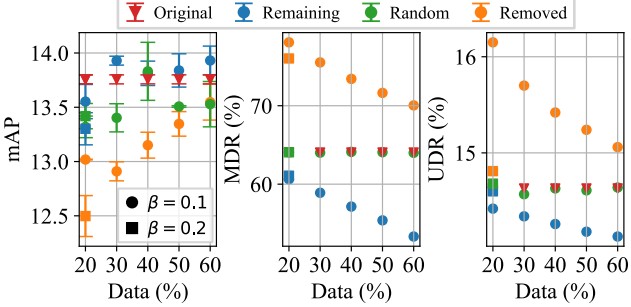

Figure 11: BDD (1% labeled). PLS results for $\delta_s = 0.4$ and $\beta = 0.1$ and $0.2$. Original student model vs. a student trained on: the remaining data post-selection using $D_i$, randomly selected data of equal size as the remaining data, and the removed data. Data (%) represents the percentage of removed data.

## 4.5 Combined Building Blocks

We evaluate the effectiveness of our proposed building blocks in addressing the key challenges of SSOD: class imbalance, erroneous GT labels, and low-quality pseudo-labeled images. To this end, we analyze their combined impact on both the teacher model $M_T$ and the student model $M_S(0.4)$. The per-class performance results are summarized in Table 7. GLC significantly improves the AP of $M_T$ trained with $N_{\text{GT}}$s at Level 1 by up to 12% in total, demonstrating its effectiveness in correcting noisy GT labels. Balancing the 1% labeled data of BDD via RCF or RCC further increases the AP by up to 8% in total, highlighting the

importance of addressing class imbalance. However, directly combining RCF and RCC leads to suboptimal class balancing, as all collages are designated as rare. As a result, they appear in every training batch, and their augmented nature results in unintended biases. To address this, collages should not influence RCF-based batch structuring and should instead be categorized as common, allowing them to be shuffled with the majority of the dataset.

We select RCC as the representative class-balancing method and evaluate its interaction with GLC and PLS. Removing 20% of pseudo-labeled images via $D_i$ with PLS leads to a total AP increase of up to 7%. Increasing the removal percentage to 60% results in only a 0.2% drop in mAP compared to removing only 20%, suggesting that the retained high-quality pseudo-labeled images provide the key supervisory signal for learning. The combination of PLS with RCC/RCF and GLC achieves the best results, leading to a total AP boost of up to 21%. This effectively mitigates the confirmation bias and overfitting issues observed in $M_S(0.4)$, where it otherwise underperforms $M_T$ on rare classes (2.34% drop for "motorcycle") while outperforming it on common classes (2.08% increase for "car"). By integrating our proposed building blocks, $M_S(0.4)$ transitions from underperforming $M_T$ by a total of 4.97% in AP to surpassing it with a total AP increase of 15.98%.

Table 7: BDD (1% labeled) with $N_{GTs}$ at Level 1. Impact of GLC, RCF ($\gamma_f = 20$), RCC ($\gamma_{r,\min} = 0.25$, $\gamma_{r,\max} = 0.75$), and PLS ($\beta = 0.1$, 20% removed) on per-class AP of $M_T$ (top) and $M_S(0.4)$ (bottom).

| GLC | RCF | RCC | PLS | Motor-cycle | Rider | Bicycle | Bus | Truck | Pedest-rian | Traffic Light | Traffic Sign | Car | $\Sigma$ |
|---|---|---|---|---|---|---|---|---|---|---|---|---|---|
| - | - | - | - | 2.02 | 6.16 | 8.12 | 21.67 | 18.85 | 15.01 | 7.07 | 12.49 | 31.23 | |
| ✓ | - | - | - | ↑2.55 | ↓2.09 | ↑1.19 | ↑2.13 | ↑3.00 | ↑0.99 | ↑**0.25** | ↑1.16 | ↑3.08 | ↑12.26 |
| ✓ | ✓ | - | - | ↑**4.38** | ↑2.04 | ↑1.07 | ↑**2.65** | ↑**3.79** | ↑1.35 | ↓0.23 | ↑**1.37** | ↑3.16 | ↑**20.04** |
| ✓ | - | ✓ | - | ↑4.13 | ↑**2.25** | ↑**1.74** | ↑2.60 | ↑2.63 | ↑1.04 | ↓0.02 | ↑1.29 | ↑**3.33** | ↑18.99 |
| ✓ | ✓ | ✓ | - | ↑2.85 | ↑1.78 | ↑1.09 | ↓1.91 | ↑1.50 | ↑0.73 | ↓0.27 | ↑0.33 | ↑2.74 | ↑8.84 |
| - | - | - | - | 2.82 | 3.82 | 6.43 | 20.86 | 17.78 | 11.48 | 7.76 | 13.39 | 33.31 | |
| ✓ | - | - | - | ↑0.02 | ↑1.87 | ↑0.51 | ↓0.53 | ↑**5.00** | ↑1.05 | ↑0.19 | ↑**1.32** | ↑2.29 | ↑11.72 |
| ✓ | - | ✓ | - | ↑1.85 | ↑1.70 | ↑0.98 | ↑1.59 | ↑3.50 | ↑1.62 | ↑0.29 | ↓0.13 | ↑2.07 | ↑13.47 |
| ✓ | - | ✓ | ✓ | ↑**2.49** | ↑**2.40** | ↑**1.88** | ↑**3.04** | ↑4.45 | ↑**2.23** | ↑**0.55** | ↑1.44 | ↑**2.47** | ↑**20.95** |

*Summary:* The combination of GLC, RCC/RCF, and PLS results in a cumulative AP boost of up to 21% in total across all classes, highlighting the necessity of a structured and dataset-aware combination of class balancing, label correction, and pseudo-label filtering in SSOD.

## 4.6 Ablation Studies

The following ablation studies present empirical validations for our parameter selection and the robustness of our methods, alongside a comparison between the selected SSOD framework STAC and other SSL and active learning (AL) frameworks.

**SSOD Framework.** We compare the basic student-teacher SSL framework to consistency-based SSL, active learning (AL), and uncertainty-based pseudo-label filtering methods. For consistency-based SSL, we re-implement CSD (Jeong et al., 2019) for EfficientDet. For uncertainty-based methods, we employ Loss Attenuation (Kendall & Gal, 2017; Kassem Sbeyti et al., 2023) and 2D spatial Monte Carlo (MC) dropout (Tompson et al., 2015) with a dropout rate of 0.05 selected based on best performance and 10 MC samples.

AL inherently addresses class imbalance by selecting uncertain, often underrepresented samples. For instance, it increases "person sitting" from 51 to 164 samples in $\mathcal{D}_{labeled}$ at the first iteration (5% start and increase to 10%) compared to random 10% sampling. The increased representation results in an mAP boost of 5% for $M_T$ and 6% for $M_S(0.4)$ over random sampling on KITTI, despite similar pseudo-label quality ( 80% MDR, 4% UDR). This underscores the potential of combining AL with SSL and the importance of class-aware analyses and methods.

We observe a consistent increase in mAP by 2% via SSL regardless of the filtering threshold and uncertainty metric used, including entropy (Roy et al., 2018), combined uncertainty (Kassem Sbeyti et al., 2024), and

epistemic uncertainty. This validates our focus on pseudo-label quality post-filtering and class-balancing over uncertainty-based filtering alone.

Additionally, CSD (Jeong et al., 2019) shows no improvement over $M_T$ at 10% on KITTI or BDD, suggesting that augmentations and consistency strategies that work on MS COCO (Lin et al., 2014) and PASCAL VOC (Everingham et al., 2010) do not necessarily generalize to real-world datasets with different challenges and characteristics.

**Detector Choice.** To evaluate the impact of detector choice on our methods, we employ the TensorFlow implementation of YOLOv3 (Farhadi & Redmon, 2018; Zhang, 2019) as an alternative to EfficientDet. We fine-tune a YOLOv3 model pretrained on MS COCO using 10% of KITTI for 50 epochs with an input resolution of 1024×1024, a batch size of 8, a learning rate of 0.0001, and anchors optimized via k-means clustering. All other hyperparameters remain at their default values. As shown in Table 8, RCC improves the performance of $M_T$ the most, increasing mAP by up to 3% and resulting in a total increase in AP by up to 19% across all classes. RCF improves the AP by up to 11% in total with batch structuring only (no augmentation or RW), consistent with its contribution to the performance of EfficienDet-based $M_T$ (see Table 3). Since our methods are data-centric, they generalize across different detectors. Unlike EfficientDet, YOLOv3 struggles on KITTI (achieving an mAP of only 6%). However, class balancing through RCC and RCF enhances performance across all classes, demonstrating the broad applicability of our methods.

Table 8: KITTI (10% labeled). Impact of RCF ($\gamma_f = 20$), and RCC ($\gamma_{r,\min} = 0.25$, $\gamma_{r,\max} = 0.75$) on per-class AP of YOLOV3-based $M_T$.

| RCF | RCC | AP | | | | | | | $\Sigma$ |
| --- | --- | --- | --- | --- | --- | --- | --- | --- | --- |
| | | Person Sitting | Tram | Truck | Cyclist | Van | Pedestrian | Car | |
| - | - | 1.09 | 2.47 | 3.15 | 2.05 | 2.76 | 6.98 | 27.51 | |
| ✓ | - | ▲**2.40** | ▲1.89 | ▲0.02 | ▲0.18 | ▲**1.58** | ▲0.49 | ▲4.44 | ▲11.00 |
| - | ✓ | ▲0.92 | ▲**3.03** | ▲0.03 | ▲**1.83** | ▲1.21 | ▲**2.67** | ▲**10.10** | ▲**18.87** |

**RCC.** Our ablation studies on RCC focus on the configuration of the collage (horizontal allocation setup vs. 4×4 grid setup), scale variation (SV), augmentation using RandAugment (Cubuk et al., 2020), and the cropping parameters $\gamma_{r,\min}$ and $\gamma_{r,\max}$. A horizontal allocation as visualized in Fig. 4 outperforms a grid setup as illustrated in Fig. 12 (left). The grid setup only improves the AP by 1.5% vs. 1.9% for "motorcycle", though it decreases it by 1.3% for "person sitting" and 0.2% for "rider" and "bicycle". Furthermore, incorporating different scales via SV (see Fig. 12 (right)) confuses the model, as the upscaling from cropping already introduces sufficient variation. SV leads to a smaller increase in AP for rare classes, e.g., "person sitting" (only 1.6% vs. 4.9%) and "motorcycle" (0.9% vs. 1.9%). Adding augmentations to the collages via RandAugment (Cubuk et al., 2020) also results in a smaller increase in AP, e.g., only 2% vs. 4.9% for "person sitting".

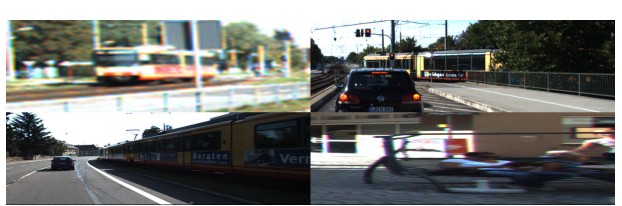 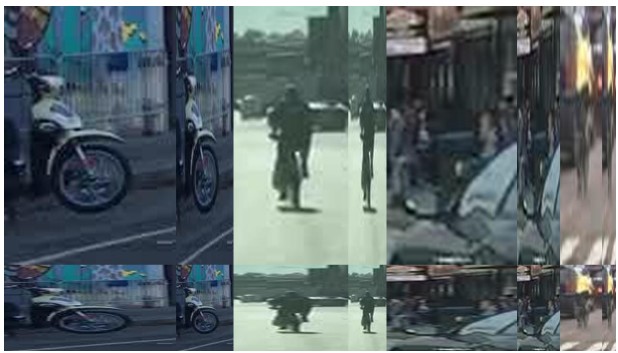

Figure 12: KITTI (left, 4×4 grid setup with $\gamma_{r,\min} = 0.5$ and $\gamma_{r,\max} = 1.0$), BDD (right, scale variation with $\gamma_{r,\min} = 0.25$ and $\gamma_{r,\max} = 0.75$). Example collages.

We demonstrate in Table 9 the robustness of our RCC method under different cropping parameters. RCC remains effective across different configurations of $\gamma_{r,\min}$ and $\gamma_{r,\max}$, with the combination of $\gamma_{r,\min} = 0.25$ and $\gamma_{r,\max} = 0.75$ yielding the highest performance on rare classes in both KITTI and BDD (up to 5%). Importantly, RCC consistently increases the performance of targeted classes without significantly impacting other classes (total increase of up to 7% across all classes ($\sum$)). While $\gamma_{r,\max} = 1.0$ results in the highest total improvements on KITTI due to a smaller drop in the AP of common classes, this comes at the cost of a smaller gain in AP for rare classes. A higher $\gamma_{r,\max}$ avoids truncating surrounding objects, maintaining performance on the common classes of the cleaner KITTI dataset, in contrst to BDD. A limitation of RCC is visible on the class "bicycle", as it does not increase the performance due to labeling inconsistencies in BDD. As shown in Table 3 and discussed in Section 4.4, the labeling of "bicycle" often includes the rider, which limits the improvement potential of the model despite the use of RCC due to inter-class confusion. This underscores the limitations of post-labeling balancing strategies.

Table 9: KITTI (top, 10% labeled), BDD (bottom, 1% labeled). Impact of $\gamma_{r,\min}$ and $\gamma_{r,\max}$ on AP of $M_{\mathrm{T}}$.

| $\gamma_{r,\min}$ | $\gamma_{r,\max}$ | Person Sitting | Tram | Truck | Cyclist | Van | Pedestrian | Car | - | - | $\sum$ |
|---|---|---|---|---|---|---|---|---|---|---|---|
| - | - | 22.89 | 43.64 | 60.14 | 45.03 | 51.70 | 36.80 | 65.99 | - | - | |
| 0.1 | 0.5 | ↑0.91 | ↑2.61 | ↓2.07 | ↓0.42 | ↑0.12 | ↓**0.19** | ↑0.07 | - | - | ↑1.03 |
| 0.25 | 0.75 | ↑**4.92** | ↑1.59 | ↓1.05 | ↓0.44 | ↓0.99 | ↓0.23 | ↓0.30 | - | - | ↑3.50 |
| 0.5 | 1.0 | ↑3.94 | ↑2.05 | ↑**1.13** | ↓0.02 | ↑0.13 | ↓0.40 | ↑0.00 | - | - | ↑**6.83** |
| 1.0 | 1.5 | ↑3.46 | ↑**3.63** | ↓0.16 | ↓0.42 | ↓0.74 | ↓0.50 | ↑0.14 | - | - | ↑5.41 |

| $\gamma_{r,\min}$ | $\gamma_{r,\max}$ | Motor-cycle | Rider | Bicycle | Bus | Truck | Pedestrian | Traffic Light | Traffic Sign | Car | $\sum$ |
|---|---|---|---|---|---|---|---|---|---|---|---|
| - | - | 4.59 | 6.66 | 9.59 | 24.72 | 23.02 | 17.07 | 7.83 | 14.71 | 35.62 | |
| 0.25 | 0.75 | ↑**1.90** | ↑**1.88** | ↓**0.29** | ↑**0.71** | ↑**0.24** | ↑**0.01** | ↑**0.09** | ↑**0.02** | ↓**0.02** | ↑**4.54** |
| 0.5 | 1.0 | ↑1.30 | ↑2.13 | ↓**0.01** | ↑0.17 | ↑0.15 | ↓0.06 | ↓0.08 | ↓0.27 | ↓0.32 | ↑3.01 |

Table 10: KITTI (top, 10% labeled), BDD (bottom, 1% labeled). Impact of $\gamma_f$ and $\mathcal{A}$ on AP of $M_{\mathrm{T}}$.

| RW | RCF | $\gamma_f$ | $\mathcal{A}$ | Person Sitting | Tram | Truck | Cyclist | Van | Pedestrian | Car | - | - | $\sum$ |
|---|---|---|---|---|---|---|---|---|---|---|---|---|---|
| - | - | - | - | 22.89 | 43.64 | 60.14 | 45.03 | 51.70 | 36.80 | 65.99 | - | - | |
| ✓ | - | 10 | - | ↓2.90 | ↑1.61 | ↓0.81 | ↑0.01 | ↑**1.30** | ↑0.45 | ↑**0.48** | - | - | ↑0.14 |
| ✓ | - | 20 | - | ↑2.72 | ↑2.64 | ↓1.08 | ↑0.26 | ↑0.76 | ↑0.32 | ↑0.31 | - | - | ↑5.93 |
| ✓ | ✓ | 10 | - | ↑2.81 | ↑1.29 | ↓2.35 | ↓0.22 | ↑0.12 | ↓0.29 | ↑0.05 | - | - | ↑1.41 |
| ✓ | ✓ | 20 | - | ↓1.33 | ↑4.14 | ↓0.38 | ↑0.16 | ↑0.15 | ↑0.27 | ↑0.14 | - | - | ↑3.15 |
| - | ✓ | 10 | - | ↑0.94 | ↑1.58 | ↑0.44 | ↓0.63 | ↓0.30 | ↑0.40 | ↓0.02 | - | - | ↑2.41 |
| - | ✓ | 20 | - | ↑2.56 | ↑0.38 | ↓0.89 | ↓1.32 | ↓0.73 | ↓0.35 | ↑0.04 | - | - | ↓0.31 |
| - | ✓ | 10 | ✓ | ↑4.47 | ↑2.58 | ↓0.79 | ↓1.14 | ↑0.61 | ↓0.37 | ↓0.36 | - | - | ↑5.00 |
| - | ✓ | 20 | ✓ | ↑3.18 | ↑4.32 | ↑1.03 | ↑0.25 | ↓0.11 | | | - | - | ↑**9.15** |
| - | ✓ | 30 | ✓ | ↑**6.14** | ↑1.27 | ↓0.53 | ↑1.02 | ↓0.63 | ↑**1.18** | ↓0.17 | - | - | ↑8.28 |
| ✓ | ✓ | 10 | ✓ | ↑1.58 | ↑**4.56** | ↑**1.40** | ↑0.65 | ↑0.43 | ↑0.22 | ↑0.00 | - | - | ↑8.84 |
| ✓ | ✓ | 20 | ✓ | ↑2.63 | ↑3.14 | ↓0.07 | ↑**1.18** | ↑0.94 | ↓0.25 | ↑0.13 | - | - | ↑7.70 |

| RW | RCF | $\gamma_f$ | $\mathcal{A}$ | Motor-cycle | Rider | Bicycle | Bus | Truck | Pedestrian | Traffic Light | Traffic Sign | Car | |
|---|---|---|---|---|---|---|---|---|---|---|---|---|---|
| - | - | - | - | 4.59 | 6.66 | 9.59 | 24.72 | 23.02 | 17.07 | 7.83 | 14.71 | 35.62 | |
| ✓ | - | 10 | - | ↓1.13 | ↑0.73 | ↓1.42 | ↓1.68 | ↓0.41 | ↓0.52 | ↓0.04 | ↑0.23 | ↓0.07 | ↓4.31 |
| ✓ | - | 20 | - | ↓0.78 | ↑0.91 | ↓1.27 | ↓2.18 | ↓0.54 | ↓0.88 | ↑0.06 | ↑0.28 | ↓0.09 | ↓4.49 |
| ✓ | ✓ | 10 | - | ↓1.87 | ↓0.16 | ↓0.33 | ↓0.59 | ↓0.32 | ↓0.31 | ↓0.03 | ↑0.23 | ↓0.13 | ↓3.51 |
| ✓ | ✓ | 20 | - | ↓0.77 | ↑1.24 | ↓1.18 | ↓1.59 | ↓0.69 | ↓0.65 | ↓0.02 | ↑0.42 | ↓**0.02** | ↓3.26 |
| - | ✓ | 10 | - | ↓0.95 | ↑1.20 | ↓0.11 | ↑**0.31** | ↓0.37 | ↑**0.28** | ↑0.12 | ↑0.09 | ↓0.19 | ↑0.38 |
| - | ✓ | 20 | - | ↓2.40 | ↑0.44 | ↓0.67 | ↓0.38 | ↑0.28 | ↑0.12 | ↑0.09 | ↑0.07 | ↓0.11 | ↓2.56 |
| - | ✓ | 10 | ✓ | ↓0.59 | ↑**2.21** | ↓1.28 | ↓0.03 | ↑**0.71** | ↑0.19 | ↓0.01 | ↑0.31 | ↓0.25 | ↑1.26 |
| - | ✓ | 20 | ✓ | ↑**1.00** | ↑1.60 | ↓**0.05** | ↓0.45 | ↑0.52 | ↑0.26 | ↑0.19 | ↑0.29 | ↑0.20 | ↑**3.16** |
| ✓ | ✓ | 10 | ✓ | ↑**1.00** | ↑1.40 | ↓0.85 | ↓0.72 | ↑0.11 | ↓0.46 | ↑0.09 | ↑0.32 | ↓0.29 | ↑0.38 |
| ✓ | ✓ | 20 | ✓ | ↓0.78 | ↑0.83 | ↓1.82 | ↓1.77 | ↑0.04 | ↓1.19 | ↑**0.25** | ↑**0.49** | ↓0.23 | ↓4.18 |

**RFC.** Our ablation studies on RFC examine the effect of different scaling strengths via $\gamma_f$ for re-weighting (RW) and RCF, both with and without augmentation $\mathcal{A}$, as well as their combined use. Depending on the selected $\gamma_f$, RW improves the AP by up to 2%, but also decreases it by up to 3% on certain classes. In contrast, RCF is less dependent on $\gamma_f$ as shown in Tables 10 and 11, with a consistent increase in performance almost across all classes, and particularly on rare classes (up to 6% in AP). Evaluating $\gamma_f = 5, 10, 20, 100$ reveals that $\gamma_f = 20$ yields the highest improvement on both datasets for $M_T$. $\gamma_f = 10$ is however sufficient for $M_S(0.4)$, as $\mathcal{D}_{pseudo}$ compensates for weaker scaling by inherently enhancing the representation of rare classes. Values of $\gamma_f$ greater than 20 only increase the mAP by up to 0.5% ($\gamma_f = 100$ on KITTI with 15% labeled). Moreover, using logarithmic smoothing in Eq. (2) further stabilizes the improvements, with an additional 1% increase in mAP compared to linear weighting at $\gamma_f = 10$. The combined use of RW and RCF proves beneficial for $M_S(0.4)$ but not $M_T$, as the additional loss term $\ell_{pseudo}$ in Eq. (1) reduces sensitivity to weight fluctuations and improves overall training stability. RandAugment (Cubuk et al., 2020) in RCF improves the AP by up to 3% due to increasing the insufficient variability in the dataset for rare classes.

Table 11: KITTI (top, 10% labeled), BDD (bottom, 1% labeled). Impact of $\gamma_f$ and $\mathcal{A}$ on AP of $M_S(0.4)$.

| RW | RCF | $\gamma_f$ | $\mathcal{A}$ | Person Sitting | Tram | Truck | Cyclist | Van | Pedestrian | Car | - | - | $\Sigma$ |
|---|---|---|---|---|---|---|---|---|---|---|---|---|---|
| - | - | - | - | 25.05 | 47.60 | 61.21 | 47.36 | 51.60 | 37.45 | 67.6 | - | - | |
| ✓ | - | 10 | - | ↑0.65 | ↑2.33 | ↑0.74 | ↓1.13 | ↑1.19 | ↓0.79 | ↓0.96 | - | - | ↑2.03 |
| ✓ | - | 20 | - | ↓2.50 | ↑2.54 | ↓0.69 | ↓1.21 | ↑0.52 | ↓0.93 | ↓1.45 | - | - | ↓3.72 |
| ✓ | ✓ | 10 | - | ↑1.15 | ↑3.71 | ↓0.73 | ↓0.65 | ↓0.26 | ↑0.63 | ↑**0.13** | - | - | ↑3.98 |
| ✓ | ✓ | 20 | - | ↓0.58 | ↑3.35 | ↓0.03 | ↑**1.03** | ↑**2.57** | ↑0.49 | ↑0.10 | - | - | ↑6.93 |
| - | ✓ | 10 | - | ↓1.98 | ↑3.24 | ↑0.71 | ↓0.06 | ↑0.36 | ↑**1.87** | ↓0.20 | - | - | ↑3.94 |
| - | ✓ | 20 | - | ↑1.39 | ↑2.47 | ↓0.09 | ↓0.39 | ↑1.44 | ↑1.30 | ↑**0.13** | - | - | ↑6.25 |
| - | ✓ | 10 | ✓ | ↑**5.15** | ↑**6.12** | ↑**3.39** | ↓0.02 | ↑1.70 | ↑1.34 | ↓0.10 | - | - | ↑**17.58** |
| - | ✓ | 20 | ✓ | ↑3.26 | ↑5.03 | ↑2.13 | ↓0.34 | ↓0.01 | ↑1.31 | ↓0.09 | - | - | ↑11.29 |
| - | ✓ | 30 | ✓ | ↑3.07 | ↑2.96 | ↑0.47 | ↓0.84 | ↑1.09 | ↑1.69 | ↓0.04 | - | - | ↑8.40 |
| ✓ | ✓ | 10 | ✓ | ↑1.56 | ↑4.16 | ↑1.79 | ↑0.63 | ↑2.42 | ↑1.06 | ↓0.32 | - | - | ↑10.04 |
| ✓ | ✓ | 20 | ✓ | ↓0.02 | ↑4.53 | ↑0.75 | ↑0.32 | ↑2.30 | ↑0.12 | ↓0.21 | - | - | ↑7.79 |

| | | | | Motor-cycle | Rider | Bicycle | Bus | Truck | Pedest-rian | Traffic Light | Traffic Sign | Car | |
|---|---|---|---|---|---|---|---|---|---|---|---|---|---|
| - | - | - | - | 4.13 | 5.62 | 7.48 | 23.89 | 22.78 | 13.63 | 9.14 | 15.37 | 36.55 | |
| ✓ | - | 10 | - | ↓0.61 | ↑1.01 | ↓0.37 | ↓0.80 | ↑0.19 | ↓0.42 | ↓0.19 | ↓0.22 | ↓0.38 | ↓1.79 |
| ✓ | - | 20 | - | ↓0.75 | ↓0.32 | ↓1.64 | ↓2.40 | ↓1.90 | ↓0.89 | ↓0.22 | ↓0.20 | ↓0.56 | ↓8.88 |
| ✓ | ✓ | 10 | - | ↓0.31 | ↑0.63 | ↑0.31 | ↑0.57 | ↓0.11 | ↑0.74 | ↑0.06 | ↑0.33 | ↓0.03 | ↑2.19 |
| ✓ | ✓ | 20 | - | ↑0.10 | ↑0.19 | ↓1.53 | ↓0.9 | ↓1.87 | ↓0.13 | ↑**0.10** | ↓0.11 | ↓0.21 | ↓4.36 |
| - | ✓ | 10 | - | ↑0.79 | ↑1.56 | ↑**1.13** | ↑0.31 | ↑0.22 | ↑1.50 | ↓0.38 | ↑**0.40** | ↑**0.19** | ↑**5.72** |
| - | ✓ | 20 | - | ↓1.34 | ↑1.14 | ↑0.75 | ↑**0.51** | ↑0.63 | ↑1.42 | ↓0.46 | ↑0.32 | ↑0.06 | ↑3.03 |
| - | ✓ | 10 | ✓ | ↓0.73 | ↑1.86 | ↑0.20 | ↑0.50 | ↑0.51 | ↑1.89 | ↓0.11 | ↑0.13 | ↓0.05 | ↑4.20 |
| - | ✓ | 20 | ✓ | ↑1.43 | ↑0.48 | ↑0.73 | ↓0.02 | ↑**0.87** | ↑**2.17** | ↓0.62 | ↑0.02 | ↓0.24 | ↑4.82 |
| ✓ | ✓ | 10 | ✓ | ↑**1.78** | ↑1.76 | ↑1.04 | ↓1.01 | ↑0.25 | ↑1.76 | ↓0.02 | ↑0.14 | ↓0.19 | ↑5.51 |
| ✓ | ✓ | 20 | ✓ | ↑0.79 | ↑**2.36** | ↓0.38 | ↓0.80 | ↑0.20 | ↑1.03 | ↓0.06 | ↑0.05 | ↓0.29 | ↑2.90 |

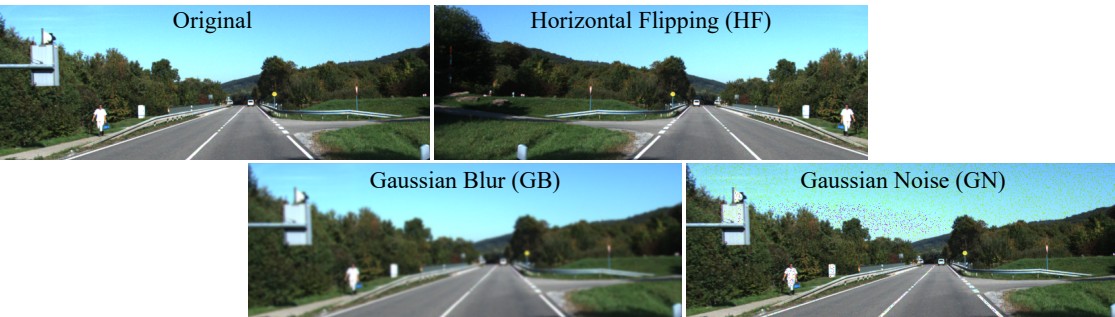

Figure 13: Augmentations applied during inference on the training set of KITTI via GLC.

**GLC.** Our ablation studies on GLC explore the impact of IoU threshold $\gamma_o$ for GT and prediction overlap, $\gamma_c$ for consistency across augmented predictions, and choice of augmentation $\mathcal{A}$ (see Fig. 13). Gaussian blur is applied using a $9 \times 9$ kernel size, and Gaussian noise is drawn with a mean of 0 and variance of 0.5.

Table 12 presents the impact of each parameter on the effectiveness of our GLC method in correcting synthetically introduced $N_{GT}$s at Level 1 to KITTI (10% labeled). The results indicate robustness to different choices of $\mathcal{A}$ and IoU thresholds, with consistent mAP recovery from added $N_{GT}$s by up to 7%. This robustness can be attributed to the use of Gaussian Noise (GN), Horizontal Flipping (HF), or Gaussian Blur (GB) solely during inference, introducing sufficient variability to evaluate model consistency effectively. GLC achieves strong performance ($47.16 \pm 0.43$ on original $\mathcal{D}_{labeled}$, $39.38 \pm 0.21$ post-corruption, and up to $46.47 \pm 0.56$ post-GLC) even with a single augmentation, thereby avoiding the additional computational cost incurred by multiple augmentations.

A similar analysis for correcting $M_{GT}$s demonstrates consistent robustness across different $\gamma_c$ values, yielding mAPs of $43.58 \pm 0.01$ for $\gamma_c = 70$, $44.08 \pm 0.06$ for $\gamma_c = 80$, and $43.88 \pm 0.02$ for $\gamma_c = 90$ (refer to Table 6).

Table 12: Impact of $\mathcal{A}$ – Gaussian Noise (GN), horizontal flipping (HF), Gaussian Blur (GB) – and IoU thresholds $\gamma_o$ and $\gamma_c$ on the efficacy of GLC in correcting added $N_{GT}$s at Level 1 to KITTI (10% labeled).

| $\mathcal{A}$ | | | $\gamma_o$ | $\gamma_c$ | mAP |
|---|---|---|---|---|---|
| GN | HF | GB | | | |
| ✓ | - | - | 90 | 90 | $46.47 \pm 0.56$ |
| - | ✓ | - | 90 | 90 | $45.99 \pm 0.72$ |
| - | - | ✓ | 90 | 90 | $45.11 \pm 0.27$ |
| ✓ | ✓ | ✓ | 90 | 90 | $45.65 \pm 0.32$ |
| ✓ | ✓ | ✓ | 80 | 90 | $45.41 \pm 0.53$ |
| ✓ | ✓ | ✓ | 70 | 90 | $45.40 \pm 0.04$ |
| ✓ | ✓ | ✓ | 90 | 80 | $45.85 \pm 0.03$ |
| ✓ | ✓ | ✓ | 90 | 70 | $44.97 \pm 0.21$ |

Furthermore, we also evaluate the importance of evaluating model consistency, compared to simply using all detections with a confidence score above the default $\delta_s = 0.4$. Relying solely on $\delta_s$ results in a reduction of up to 4% in mAP, achieving only an mAP of $41.32 \pm 0.17$ post-correction of $N_{GT}$s at Level 1 on 10% of KITTI.

Our experiments on $N_{GT}$s focus on box perturbations, as we assume that noisy ground truth labels primarily arise from inaccuracies in box locations rather than misclassification of objects. Manual labeling often struggles with precise box placement but rarely misidentifies object categories. For instance, confusing a car with a pedestrian is less likely than an imprecise bounding box placement. This assumption is supported by the high mACC and lower mIoU of the detector on both datasets (see Table 1), as well as by previous work demonstrating that spatial inaccuracies in labeling can significantly degrade model performance (Grad et al., 2024).

On KITTI, the detector encounters difficulties in crowded scenes. While the overlapping nature of objects in such scenes complicates boundary identification, the primary issue arises from low-quality box labels, as visualized in Fig. 14.

Nevertheless, to investigate the effect of class label noise, we randomly mislabel 20% of the objects in KITTI (10% labeled), corresponding to level 1 $N_{GT}$. The 20% noise result in a reduced label accuracy of 86.83%. Applying our consistency-based correction method, where the class labels are corrected using predictions that remain consistent across multiple inferences under data augmentations, increases label accuracy from 86.83% to 95.70%. This results in an mAP improvement from $35.91 \pm 0.22$ under

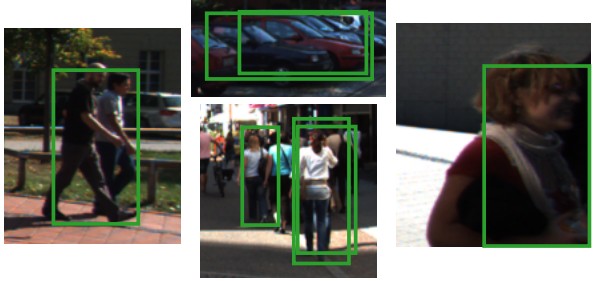

Figure 14: Examples of noisy box labels in KITTI. Human annotators often fail to accurately label occluded and truncated objects, especially in crowded scenes.

noise to $42.55 \pm 0.14$. Despite this correction, the approximately 4% remaining loss in class label accuracy demonstrates the sensitivity of the model to it, as the original labels yield a substantially higher mAP of 47.16% (see Table 6).

Label noise, whether in class or box labels, appears to have a comparable impact on model performance. In the absence of significant label errors (see Section 4.4 and Table 5), the primary bottleneck in SSOD

lies in class distribution and object availability. This highlights the critical importance of addressing class imbalance to achieve substantial improvements in object detectors.

**PLS.** Our first ablation study on PLS explores the correlation between $M_D$s and our $S_i$ metric at varying $\delta_s$. Increasing $\delta_s$ results in a higher proportion of $M_D$s, with $S_i$ showing a proportional increase (see Fig. 15), thus validating the effectiveness of our metric. Next, we assess the impact of $\beta$ in Eq. (3) on the class distribution of removed pseudo-labeled images at 30% of KITTI labeled. Fig. 16 illustrates the shift in class distribution for different $\beta$ values via the relative class count calculated as $\left( \frac{f_{k_{D_i(\beta)}} - f_{k_{S_i}}}{f_{k_{S_i}}} \right) \cdot 100$. A higher $\beta$ retains more rare classes, with the trade-off of a weaker correlation with $M_D$s. Compared to $\beta = 0.1$, $\beta = 0.25$ retains 95% more of the rare class "person sitting", while removing 70% more of the most common class "car". Despite these adjustments, the AUC only decreases by 10%, as shown in Fig. 9, highlighting the robustness of Eq. (3). Identical behavior is observed on BDD.

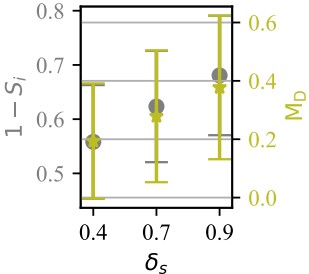

Figure 15: Correlation between our metric $S_i$ and the proportion of missing detections $M_D$s per image for varying values of $\delta_s$ on KITTI.

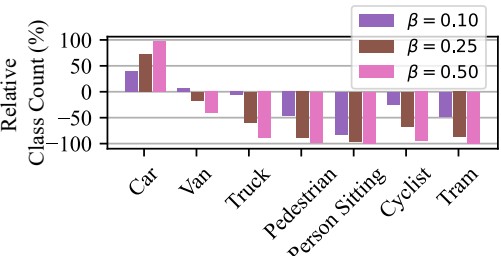

Figure 16: Impact of $\beta$ on the relative class count in the removed pseudo-labeled images at 30% of KITTI labeled with $\delta_s = 0.4$.

## 5 Conclusion

We conduct extensive analyses on label quality and its impact on SSOD performance. Our findings indicate that effective pseudo-labels-based SSOD requires a fundamental understanding of the challenging factors affecting label quality, such as class distribution, noise, and the precision-recall trade-off. By addressing dataset imbalances and implementing data-centric quality measures, we demonstrate that even basic SSOD frameworks can perform well under real-world conditions. We propose four novel, computationally efficient, and broadly applicable building blocks to improve the effectiveness SSOD. Our building blocks consist of two methods for balancing training data to increase rare class representation (RCC, RCF), a method to enhance labeled data quality (GLC), and a method to refine the selection of pseudo-labeled images (PLS). Through comprehensive experiments across different data configurations on the KITTI and BDD100K autonomous driving datasets, we show that our methods significantly improve the effectiveness of student-teacher SSOD frameworks when integrated individually. Our results highlight the significant potential of our methods and pave the way for future research in SSL.

## 6 Discussion and Future Work

We evaluate each proposed method individually to isolate its specific impact on SSOD, as joint evaluation risks conflating their contributions. Therefore, we focus on a detailed analysis within a controlled setting, providing a clear foundation for future work to build upon and explore these methods in combination and across other application domains. For instance, in the context of filtering in the SSL framework, combining more advanced filtering methods, such as those proposed by (Chen et al., 2022; Li et al., 2023; Kimhi et al., 2024), and training strategies, including those by (Li et al., 2022a; 2023), with our building blocks holds potential for more significant improvements. Our experiments demonstrate that combining different

uncertainties (epistemic and aleatoric) achieves a 41% MDR at a 1% UDR, compared to 46% MDR at the same UDR with $\delta_s$.

Understanding the influence of training parameters on detector performance and robustness, particularly in the presence of noise, is a promising direction for future work. Additionally, examining how robustness evolves across different stages of training could offer valuable insights into optimizing SSL frameworks. Given the additional hyperparameters introduced by each building block, reducing sensitivity to hyperparameter selection would enhance the practical usability of our approach. Automating batch balancing in RCF, optimizing the removal ratio in PLS, and refining consistency thresholds in GLC through hyperparameter tuning techniques or automated optimization methods could help identify more generalizable configurations.

Our building blocks are designed to be as model- and framework-agnostic as possible, avoiding reliance on domain-specific assumptions. They utilize available labels and predictions without requiring modifications to the underlying detector architecture or SSOD framework. RCC generates collages based on ground truth labels. GLC refines ground truth labels using teacher predictions and augmentation consistency, assuming the availability of a trained teacher detector. RCF only requires changes to the dataloader for batch structuring. PLS assumes a trained teacher detector that outputs confidence scores and incorporates filtering as part of its post-processing, a common feature in most object detectors but not universal. This highlights the advantages of evaluating with additional detectors, such as Transformer-based models, as part of future work. Our results demonstrate that RCC, RCF, and GLC improve teacher performance, with benefits extending to the student, particularly through PLS. However, since our empirical validation involves a single detector type (EfficientDet-D0 and YOLOv3) and a single SSOD framework (STAC), additional evaluation across diverse detectors and SSOD frameworks would strengthen the potential for widespread adoption of our methods. KITTI and BDD provide diverse and challenging conditions reflective of real-world datasets, including class imbalance, label noise, and varying dataset sizes. Extending the evaluation to other domains, such as medical imaging, satellite imagery, and industrial quality control, would further demonstrate the broad applicability and robustness of our methods.

PLS effectively identifies low-quality pseudo-labeled images post-filtering. However, these are currently discarded. Integrating AL with SSL could enhance $M_S$ performance by utilizing those images. Low-quality pseudo-labeled images could be directed either to an oracle or included in a consistency-based unsupervised loss, enabling further feature extraction in a self-supervised manner. Moreover, adding depth estimation to our metric $D_i$ could improve its correlation with missing detections and reduce reliance on teacher model score quality, as distant objects often have higher miss rates than nearby ones.

Curriculum learning approaches could also be explored to complement SSL and AL. Specifically, we propose defining a curriculum for $M_S$ based on the learning progression of $M_T$ to mitigate error propagation. For example, adopting a ramp-up strategy by starting with labeled data before gradually incorporating pseudo-labeled data might outperform using both from the start. Moreover, alternative weighting strategies for ranking images in RCF, considering various forms of prior knowledge beyond class frequency, could be further investigated.

Constructing collages of rare pseudo-labeled objects could also further increase performance via RCC, especially under low data regimes. However, collages introduce a trade-off between addressing class imbalance and potential distribution shifts, highlighting the possible limitation of this approach. Using the same training dataset to generate the collages ensures consistency with the original data distribution but also limits the diversity of objects and contexts available for training. Gathering collages from other datasets containing similar classes is promising despite significant challenges such as differences in labeling conventions and the semantic understanding of class definitions across datasets, as also highlighted by (Liu et al., 2022a). For example, distinctions between classes such as "van", "truck", and "car" often vary based on annotator interpretation, potentially introducing additional label noise. Moreover, matching rare classes across datasets poses significant challenges. For instance, BDD contains rare classes such as "rider", which are absent in KITTI, making cross-dataset collage generation infeasible in specific scenarios. Future work could explore methods to enable more effective collage generation in RCC by integrating domain adaptation techniques and utilizing large-scale datasets with harmonized class definitions.

For GLC, our work could be extended to analyze class-based errors. Given the excellent performance of the classification head across datasets, optimizing the combination of class and box losses could further improve the quality of pseudo-labeled samples.

In summary, basic SSOD frameworks become more nuanced when confronted with real-world data. We hope this work encourages deeper investigation into SSOD, moving beyond incremental gains toward a more comprehensive understanding of the core challenges and mechanisms underlying SSL.

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
