# OpenReview forum: "Building Blocks for Robust and Effective Semi-Supervised Real-World Object Detection"
_TMLR — Accepted by TMLR_

### Review · Reviewer_Xcrj · 2024-12-06

**Summary Of Contributions:**

The authors contribute a novel empirical study in the field of semi-supervised object detection (SSOD) for self-driving datasets. They enumerate three causes of suboptimal pseudo-labeling (class imbalance, noise in ground truth labels, incomplete or inaccurate pseudo-labels) and propose remedial methods for each of the shortcomings.

**Audience:**

Yes

**Broader Impact Concerns:**

None.

**Claims And Evidence:**

No

**Requested Changes:**

W1. The paper's claim that the methods they propose are model-agnostic and framework-agnostic is not sufficiently supported by the experiments, which consist of a single model and a single framework. It is not clear from these experiments alone that the conclusions drawn by the authors would hold true for other models and frameworks. I would request either that the authors change the claim that their methods are model-agnostic and framework-agnostic, or add another model and another framework to the experiments in the paper.

**Strengths And Weaknesses:**

The writing and presentation of this work is one of its strengths. Sections are ordered sensibly, figures and tables are presented well, and the text is very readable. The authors' choice of interventions to study are simple but also practical and useful. The experiments are reasonably comprehensive as well.

---

> ### Author Response · Authors · 2024-12-09
>
> We thank Reviewer Xcrj for the concise feedback and for highlighting the strengths of our paper, particularly its clarity, presentation, and experimental comprehensiveness. Below, we address your concern regarding the claims of model- and framework-agnostic applicability.
>
> ### Clarifying the Model- and Framework-Agnostic Claims
>
> Our claims are based on the design principles of the proposed methods **Rare Class Collage (RCC)**, **Rare Class Focus (RCF)**, **Ground Truth Label Correction (GLC)**, and **Pseudo-Label Selection (PLS)**, which are independent of specific model architectures or frameworks:
>
> - **Model-Agnostic Nature:** All our methods rely solely on labels and predictions and do not require any modification to the underlying detector architecture. For example:
>     - **RCC** generates collages based on ground truth labels.
>     - **GLC** refines ground truth labels using teacher predictions and augmentation consistency. This assumes the availability of a trained teacher detector.
>     - **RCF** only requires changes to the dataloader for the stratification process, as outlined in the paper.
>     - **PLS** assumes a trained teacher detector that outputs confidence scores and incorporates filtering as part of its post-processing. While this is a standard feature in most object detectors, we acknowledge that it does not apply universally.
>
> - **Framework-Agnostic Nature:**
>   We demonstrate in the paper that our proposed methods **RCC**, **RCF**, and **GLC** improve the performance of the teacher detector, thereby providing benefits even outside the context of semi-supervised object detection (SSOD). Additionally, we show that these improvements carry over to the student, particularly with **PLS**, which leverages standard SSOD components such as teacher-student training, the availability of pseudo-labels, and access to confidence scores. This forms the basis of our claim that the methods are independent of the specifics of the underlying SSOD framework.
>
> ### Acknowledgment and Proposed Changes
>
> We affirm and recognize the valid concern that our empirical validation involves a single model (EfficientDet-D0) and a single framework (STAC). To address this and clarify the potentially ambiguous term "model-agnostic":
>
> - We will revise the paper and frame the claim as "designed to be model- and framework-agnostic, ensuring broad applicability" while explicitly stating the assumptions and constraints underlying the methods' applicability, as described above.
> - We will consolidate the claim and its clarifications to reduce redundancy and avoid repeating it throughout the paper. A dedicated part in the discussion will outline the limitations of our evaluation. This will emphasize the advantages of broader empirical validation across diverse models (e.g., Transformer-based detectors) and SSOD frameworks as part of future work.
>
> To ensure any ongoing review process is not disrupted, we will provide a revised manuscript reflecting these clarifications once all reviews are submitted. We appreciate your insightful comments and hope these revisions will provide transparency and align our claims with the demonstrated scope of our extensive experiments.
>
> Thank you once again for your valuable and constructive feedback.

---

### Review · Reviewer_2jgj · 2025-01-03

**Summary Of Contributions:**

The paper suggest two components for SSOD paradigme, that can be sub-categorized into four, focusing on class rareity and label refinement. each module categorize the action by the definition of problem, as defined by the paper, varying from real labels to pseudo labels, missing to noisy and so on.

**Audience:**

Yes

**Broader Impact Concerns:**

No concerns raised about the broader impact

**Claims And Evidence:**

Yes

**Requested Changes:**

My main concerns fall into two categories: hypothesis validation and comparable methods-validated results.

# 1. Hypothesis Validation

* The authors make several statements, for example, claiming that noisy labels in object detection are not addressed, while citing claims about the effects of label noise from classification tasks. This is indeed an issue, for example in [1]; however, even in instance mask prediction, most of the effects on the model derive from misclassification, as presented in [2]. They argue for spatial label noise, yet the ablation shows that most of the performance hindrance comes from class noise.

* Other statements, such as the certainty approximation based on the model's confidence for a specific image, need to be validated by collecting statistics on the number of instances per image, model calibration, and different stages of training (e.g., an untrained model is more likely to be uncalibrated).

* Consider an alternative for $𝑆_i(\delta_i)$, such as the margin between the first and second class scores of the classification head [7], as it is robust to the number of instances in an image.

* You claim that the blocks are model-agnostic. I disagree, as the method builds exclusively on EfficientDet. Modern detectors, such as DETR, operate differently, and some of the assumptions may not hold when using object queries instead of anchors. Additionally, your method relies on a teacher-student paradigm that leverages teacher-student consistency. If you aim to be framework-agnostic, you would need to modify this appropriately.

* A final note on this aspect: you utilize "copy-paste augmentation" [3] to mitigate imbalanced predictions. However, this approach completely disrupts the natural image distribution. I wonder about its effects on object detection "in the wild." By creating a collage, you may address the class imbalance issue but simultaneously introduce out-of-distribution images that are unlikely to occur in real-world scenarios. It would be beneficial to see examples from a different dataset or simply outside the validation set and analyze the cost/effects of your method.


# 2. Comparable Methods

* The literature includes a wealth of SSOD approaches [4,5], including those addressing class imbalance [6]. Since your paper does not address any aspect unique to the data presented, it would be more compelling to compare your results with these existing methods. I believe it would strengthen your argument significantly if you applied your method to LVIS, where class imbalance is prominent (and where various existing papers have conducted similar experiments). While not mandatory, I think testing your imbalance correction on COCO would also be appropriate, given its larger set of classes.

* Please add "cascaded two-stage thresholding" [8] to the related work section.


### References:

[1] Towards Robust Adaptive Object Detection under Noisy Annotations

[2] Benchmarking Label Noise in Instance Segmentation: Spatial Noise Matters

[3] Simple Copy-Paste is a Strong Data Augmentation Method for Instance Segmentation

[4] Label Matching Semi-Supervised Object Detection

[5] Semi-Supervised Object Detection via Multi-instance Alignment with Global Class Prototypes

[6] Gradient-based Sampling for Class Imbalanced Semi-Supervised Object Detection

[7] Semi-Supervised Semantic Segmentation via Marginal Contextual Information

[8] Robot Instance Segmentation with Few Annotations for Grasping

**Strengths And Weaknesses:**

I particularly liked that the authors make an effort to use real time predictions statistics as a quality assesment for both the model and the data. While existing methods typically relay on more treditional robust assesments, it is usually a computational burden (spesifically for object detection, assesing the quality of the regression is extremly hard task to preform efficiently, and most methods use all data statistics or 100 pertubated boxes to asses quality).
The paper is well sectioned, clear figures and description of each module, as well as experiments are well documented.

---

> ### Author Response · Authors · 2025-01-06
>
> We sincerely thank Reviewer 2jgj for their thoughtful and constructive feedback, as well as for highlighting the strengths of our work. Your insights are invaluable and have greatly contributed to improving the rigor and scope of our study. Based on your feedback, we have conducted additional experiments and provided further clarifications to strengthen our claims.
> ## 1. Hypothesis Validation
> ### **Regarding Class and Box Noise**
> Your observation about the sensitivity of object detection and segmentation models to class label noise prompted us to conduct additional experiments. We introduced **20% class label noise** (random mislabeling of class annotations) to the KITTI dataset (10% labeled), resulting in a reduced label accuracy of **86.83%**. Our original assumption was that noisy ground truth labels primarily arise from inaccuracies in box locations rather than misclassification of objects. This is because manual labeling often struggles with precise box placement but rarely misidentifies object categories. For instance, confusing a car with a pedestrian is less likely than an imprecise bounding box placement. This assumption is supported by the high classification accuracy (mACC) of the detector on both datasets in comparison to the lower mIoU, as shown in Table 1.
>
> To address the class label noise, we applied our consistency-based correction method, where ground truth class labels are corrected using predictions that remain consistent across multiple inferences under data augmentations. This method increased label accuracy from **86.83%** to **95.70%**, resulting in an mAP improvement from **$35.91 \pm 0.22$** under noise to **$42.55 \pm 0.14$**. Despite this correction, the approximately **4%** remaining loss in class label accuracy demonstrates the model's sensitivity to class label quality, as the original ground truth labels yield a substantially higher mAP of **47.16%** (see Table 6).
>
> However, our findings suggest that label noise alone, whether in class or box annotations, does not solely explain performance degradation. Both types of noise appear to have a comparable impact on model performance, as shown in Table 6 and with the new results. Instead, the primary bottleneck lies in class distribution and object availability. This conclusion aligns with your observation, with Grad et al.‘s [2], and further underscores the necessity of addressing class imbalance to achieve robust improvements in object detection models.
>
> We will revise the paper to better contextualize the relative contributions of class and box label noise by adding our new experiments to the ablations. Additionally, we will include the work by Grad et al. [2] in the references, as it complements our findings from the domain of instance segmentation, demonstrating that spatial inaccuracies in labeling can significantly degrade model performance.
>
> ### **Validation at Different Stages of Training**
> We acknowledge the importance of investigating the impact of noise at different stages of training. While this analysis is beyond the scope of our current work, we have added a discussion in Section 6 to highlight its significance. Semi-supervised learning frameworks rely heavily on the selected training parameters, and understanding how these parameters influence performance and the detector's robustness, as well as how robustness to noise develops during training, is a promising direction for future research. We thank the reviewer for this valuable suggestion.

---

> > ### Author Response · Authors · 2025-01-06
> >
> > ### **On Using Margin Between Class Scores for MDR Estimation**
> > We explored your suggestion to estimate the missing detection rate (MDR) using the margin between the first and second class scores (inspired by Kimhi et al. [7]). However, the significant differences between segmentation and object detection make direct adaptation challenging. Two possible implementations for object detection were considered:
> >
> > 1. The margin between scores across all detections within an image.
> > 2. The margin between the first and second logits for each individual prediction, akin to measuring entropy across logits for a single prediction.
> >
> > We implemented approach (1) since the scores are readily available, and it yielded an AUC of **0.55**, compared to **0.90+** for our $ S_i $ metric (see Figure 8). The discrepancy can be attributed to fundamental differences between object detection and segmentation tasks. Object detection involves post-processing steps, such as anchor-based predictions, sigmoid activation, and NMS filtering, which discard many low-confidence detections. For example, EfficientDet generates over 50,000 anchors per image (given our selected resolution and anchor settings), each assigned a confidence score between 0 and 1 after applying sigmoid to the logits. Only a subset of these are retained after filtering via $\delta_s$. Consequently, relying on margins can be misleading: if two detections in an image receive a high confidence score (e.g., both scoring 1), the model may falsely appear to have no missing detections in the image, even though numerous low-confidence detections were discarded.
> >
> > For approach (2), comparing margins between logits for each detection would require access to the detector’s raw logits and the aggregation of margins across the image. However, this access is not automatically provided, as object detectors typically retain only the maximum predicted confidence across all classes for each detection, applying sigmoid only to these maximum scores. Accessing the raw logits would require propagating through all NMS steps. While conceptually promising, this approach introduces additional complexity. Furthermore, we expect low confidence scores in the output of the default post-processing pipeline to already indicate the detector's uncertainty for specific predictions, which would naturally correlate with smaller margins. Reliance on margins alone for each detection is less promissing in object detection because it fails to account for the overall distribution of detections within an image. We already evaluate metrics aggregating scores across detections, such as the average score per image, which perform worse than our $ S_i $ metric (see Table 8). This is because the average score does not capture the inherent variability in score distributions within an image. Object detection outputs often exhibit disparities in confidence scores among detections within a single image, where high-confidence detections overshadow lower-confidence detections. This affects exactly the instances that are difficult to detect or entirely missed, rendering metrics based solely on scores or score margins insufficient. In contrast, our $S_i$ metric evaluates the number of detections at both low and high filtering thresholds, offering a more robust measure of detection quality by incorporating variations in detection counts at the different score thresholds. This approach effectively captures factors such as image complexity and object density. For instance, on KITTI, the detector struggles with crowded scenes where overlapping objects lead to occlusions and reduced confidence, while on BDD, it faces challenges with distant, densely packed objects that are harder to distinguish. These challenges directly influence detection scores and the likelihood of missing detections, which is reflected in the number of detections at different $\delta_s$ values and effectively captured via our $S_i$ metric. For instance, if the detector produces one detection at $\delta_s = 0.9$ but 200 detections at $\delta_s = 0.1$, it is more likely that the image contains missing detections compared to a scenario where the detector has only five detections at $\delta_s = 0.1$. We hope this explanation clarifies the intuition behind our approach.

---

> > > ### Author Response · Authors · 2025-01-06
> > >
> > > ### **Model-Agnostic and Framework-Agnostic Claims**
> > > We appreciate your thoughtful feedback on the model-agnostic claims and agree that certain assumptions underlying our building blocks do not generalize to all detection architectures, such as DETR. We revised the paper to clarify the dependencies of each block and explicitly state the limitations of their applicability (see response to Reviewer Xcrj). Thank you for emphasizing the importance of accurate phrasing, as it allowed us to refine our claims and provide a clearer scope for our contributions.
> > >
> > > ### **Copy-Paste Augmentation and Out-of-Distribution Generalization**
> > > We agree that using collages introduces a trade-off between addressing class imbalance and potential distribution shifts. To clarify, our collages are created using the same training dataset, ensuring in-distribution augmentation. However, we acknowledge the limitations of this approach when applied to "in-the-wild" scenarios. Currently, the scope of the paper assumes in-distribution data, as we use only the same training dataset to generate the collages. This ensures consistency with the original data distribution but also limits the diversity of objects and contexts available for training.
> > >
> > > While gathering collages from other datasets containing similar classes may seem promising, such an approach presents significant challenges. These include differences in labeling conventions and the semantic understanding of class definitions across datasets, as also highlighted by Liu et al. [1]. For example, distinctions between categories like van, truck, and car often vary based on annotator interpretation, potentially introducing additional label noise. Additionally, matching rare classes across datasets is inherently difficult. For instance, the BDD dataset contains rare classes such as "rider," which are absent in KITTI, making cross-dataset collage generation infeasible for some scenarios.
> > >
> > > Despite these challenges, we believe this remains a valuable future direction. Exploring methods to mitigate labeling inconsistencies, incorporating domain adaptation techniques, or leveraging large-scale datasets with aligned class definitions could potentially enable effective out-of-distribution collage generation. We are excited to explore these possibilities in future work and have added this to the discussion in Section 6.
> > >
> > > Furthermore, regarding copy-paste augmentation, while Ghiasi et al. [3] demonstrate its effectiveness for segmentation, object detection labels are rectangles that include background along with the object. This contrasts with instance segmentation, where objects can be seamlessly integrated into new images. We have clarified this distinction in the revised manuscript. We sincerely thank you for highlighting these important considerations and for providing valuable suggestions that have helped us refine the scope and limitations of our method.

---

> > > > ### Author Response · Authors · 2025-01-06
> > > >
> > > > ## 2. Comparable Methods
> > > > We sincerely thank you for bringing these references to our attention and for highlighting their relevance to our work. Below, we provide a detailed explanation of how our approach relates to these methods and how they complement or differ from our contributions.
> > > >
> > > > - **Label Matching Semi-Supervised Object Detection [4]**:
> > > >   This work introduces a novel self-training framework that modifies the teacher-student architecture by injecting student proposals into the teacher to generate accurate pseudo-labels. It also employs adaptive label-distribution-aware confidence thresholds to refine pseudo-labels dynamically. While this approach addresses the generation and refinement of pseudo-labels, our method, particularly PLS, focuses on handling missing detections post-thresholding. Thus, these approaches are complementary rather than directly comparable. We will add their adaptive thresholding mechanism to the related work section under dynamic thresholding, emphasizing that our PLS can be used in conjunction with such methods to estimate and mitigate the impact of missing detections that persist despite advanced thresholding strategies.
> > > >
> > > > - **Semi-Supervised Object Detection via Multi-instance Alignment with Global Class Prototypes [5]**:
> > > >   This work proposes enhancing prediction consistency using global class prototypes. The framework enforces consistency regularization and introduces a loss function to improve pseudo-labels. While this method focuses on improving pseudo-label quality through prototype-based consistency regularization, our approach can complement it by addressing missing detections and rebalancing rare classes. We will include this reference in the related work section and clarify its alignment with our contributions.
> > > >
> > > > - **Gradient-based Sampling for Class Imbalanced Semi-Supervised Object Detection [6]**:
> > > >   This framework introduces methods to address class imbalance in SSOD. They dynamically adjust weights and thresholds based on class-wise gradients and confidence levels to obtain class-balanced pseudo-labels. Our methods, such as RCC and RCF, address class imbalance differently by directly rebalancing the training data rather than modifying loss functions or pseudo-label sampling. Furthermore, our PLS focuses on identifying and excluding images with persistent missing detections, which is orthogonal to their approach. While their method, Gradient-based Thresholding, aligns with our dynamic thresholding discussions, we reiterate that our methods are plug-and-play and do not require modifications to the model architecture or loss functions. We will add this reference to the related work section to position their contributions alongside our methods effectively.
> > > >
> > > > While these three works are indeed relevant and could be incorporated into our methods to explore complementary benefits, as mentioned in the discussion, we leave such integrations and comparisons to future work due to the scope of this paper. Our primary focus is on investigating the specific challenges of failure modes in SSOD and proposing modular methods that directly address these issues. Nevertheless, we acknowledge the contributions of these works and have added citations and discussions for all the suggested references (Chen et al. [4], Li et al. [5], Li et al. [6]) in the related work and discussion sections to ensure our methods are effectively positioned within the broader context of SSOD research. We sincerely thank you for pointing out these references, which have helped us strengthen the context and relevance of our work.

---

> > > > > ### Author Response · Authors · 2025-01-06
> > > > >
> > > > > ### **Experiments on COCO and LVIS**
> > > > > We appreciate your thoughtful comment and the suggestion to explore datasets like LVIS and COCO. We chose autonomous driving datasets (KITTI and BDD) as our primary use-case due to their real-world relevance and the unique combination of challenges they present, such as rare classes, occluded objects, and noisy labels. While extending our experiments to LVIS and COCO would indeed strengthen the generalizability of our methods, these datasets present a broader set of challenges, including semantic segmentation tasks in the case of LVIS, that are beyond the scope of the current paper. We acknowledge the importance of testing our imbalance correction methods on more diverse datasets, as we already mention in the discussion (see Section 6), particularly those with pronounced class imbalance such as LVIS. We will consider this as a valuable direction for future research. Your suggestion has highlighted a key opportunity to expand the applicability of our methods, and we sincerely thank you for this constructive feedback.
> > > > >
> > > > >
> > > > >
> > > > > ### **Adding Cascaded Two-Stage Thresholding [8]**
> > > > > We have added "cascaded two-stage thresholding" from the work by Kimhi et al. [8] to the related work section and acknowledged its relevance in refining pseudo-labels during training. This method complements the dynamic thresholding discussion in the related work and improves the comprehensiveness of our references. Thank you for the suggestion. We particularly appreciate the integration of time-variability in this approach, which allows the threshold to adapt and relax as the model gains confidence during training.
> > > > >
> > > > > ## Summary
> > > > > We deeply appreciate your insightful comments and the opportunity to strengthen our paper. Your feedback has led to:
> > > > > - **Additional experiments** (e.g., class noise ablation and margin-based MDR estimation).
> > > > > - **Clarifications** on limitations and future directions.
> > > > > - **Expanded related work** with added references and discussions.
> > > > >
> > > > > We hope these updates address your concerns and further demonstrate the robustness of our contributions. Thank you once again for your thoughtful review. Once all three reviews are uploaded, we will submit a revised version incorporating all the changes mentioned to ensure the reviewing process proceeds smoothly.

---

> > > > ### Comment · Reviewer_2jgj · 2025-01-07
> > > > **Revision**
> > > >
> > > > please provide the revised version

---

> > ### Comment · Reviewer_2jgj · 2025-01-07
> > **Class and Box Noise**
> >
> > I apologize that my comment was not clear, my intention was not to add class noise, but that your assumptions on noise in data step in research that talk about class noise. I fully agree with the statement that most noise in annotation derrive from noisy bounderies.
> >
> > Despite not being my intention, it seems like an interesting experiment that led to an insight about the robustness to noise, so it was not a waist of time.
> >
> > My main concern is about correction of assumptions, even if you observe empirically noise in boxes in GT, you can give qualitevly examples in the appendix and support your claim, but as for now, the support relay on research that argue for image classification label noise.

---

> > > ### Author Response · Authors · 2025-01-07
> > >
> > > Thank you for the constructive feedback and for clarifying your earlier comment. To support our claim about box label noise and its impact on model performance, we have added a new figure (Fig. 13 in the revised paper) in the ablation section under GLC. This figure visualizes examples of box label inaccuracies in the KITTI dataset, specifically highlighting how the model struggles in crowded scenes due to poor-quality annotations. The visualization underscores that the challenges are not only due to the inherent difficulty of detecting objects in crowded environments but also because of the limitations in the ground truth annotations, particularly in scenarios involving occluded objects and overlapping boundaries.
> > >
> > > We hope this addition provides clearer qualitative support for our observations and strengthens the empirical grounding of our conclusions. Thank you again for your valuable suggestions, which have greatly enhanced the clarity and rigor of our work.

---

### Review · Reviewer_h5gt · 2025-01-19

**Summary Of Contributions:**

The paper first provides an in-depth analysis of pseudo-labeling-based Semi-supervised Object Detection (SSOD) to identify the potential causes of suboptmial performance in SSOD. Next, it proposes four building blocks that can be incorporated into SSOD frameworks: Rare Class Collage (RCC), Rare Class Focus (RCF), Ground Truth Label Correction (GLC), and Pseudo-Label Selection (PLS). Finally, they experimentally demonstrate that using the proposed building block can enhance the performance in the SSOD task.

**Audience:**

Yes

**Claims And Evidence:**

Yes

**Requested Changes:**

Major changes:
1) The authors should provide an experiment to show the performance gain from incorporting all proposed building blocks together (or mention where the corresponding results can be found if I missed it)

2) Additional model should be used in the evaluations. Due to the volume of the experiments, only a small subset of the experiments can be conducted with the new model.

Minor change:
1) Considering Pseudo-Label Selection (PLS) as metric is confusing. I think the paper should refer to PLS as a method instead of a metric.

**Strengths And Weaknesses:**

Strengths
1) Well-written and well-presented
2) Extensive experiments to show the effectiveness of the proposed components
3) Useful discussion on the limitations of the work


Weaknesses
1) The collective performance gain from all four components seems to be missing (or maybe I missed it)
2) A single model (EfficientDet-D0) has been employed in the evaluations
3) The performance depends on carefully choosing many hyper-parameters/thresholds (e.g. threshold for rare/common images, confidence score threshold, and etc)

---

> ### Author Response · Authors · 2025-01-31
>
> We sincerely thank Reviewer h5gt for their accurate summary of our contributions and for highlighting the strengths of our work. We also greatly appreciate the thoughtful feedback and constructive suggestions, which allow us to further refine and improve our paper. Below, we address each point in detail.
>
> ## 1. Performance Gain from Combining All Building Blocks:
> In our work, we focus on isolating and analyzing the individual impact of each identified challenge in SSOD and the corresponding proposed method in a controlled environment. The methods themselves do not conflict with each other. For example, using GLC to clean the ground truth does not interfere with the performance boost provided by RCC for class balancing or PLS for selecting pseudo-labeled images. Thus, we analyze each building block separately to provide a detailed understanding of their contributions.
>
> However, in response to your valuable suggestion, we are conducting experiments to evaluate the performance gain from combining all proposed methods, offering an overview of their joint contribution to overall performance. We will include the results in Section 4 of the revised paper, which we will submit in the coming days once the experiments are completed (training the student model $M_{\mbox{S}}$ on BDD with an NVIDIA A100 GPU takes more than 10 days with the selected input resolution).
>
> ## 2. Hyperparameters:
> We acknowledge your concern regarding the number of hyperparameters required for each building block. Below is a summary of the hyperparameters associated with our methods:
>
> 1. **RCC**: Selection of target classes and cropping parameters for the collage.
> 2. **RCF**: The number of rare examples per batch. When combined with RW, the maximum weight assigned to the images.
> 3. **GLC**: The IoU thresholds for determining detection consistency and associating detections with their ground truth.
> 4. **PLS**: The value of $\beta$ and the proportion of images to be removed.
>
> While we recognize that hyperparameters may be perceived as a limitation, they are currently unavoidable at this stage of our work. However, our ablation studies in Section 4.5 provide a detailed evaluation of the robustness of our building blocks to these hyperparameters, offering practical guidelines for their selection and suggesting reasonable default values to facilitate adoption without extensive tuning. This is further demonstrated by our new experiments using YOLOv3 as a detector, as per your request. Additionally, we will expand the discussion in Section 6 to explore potential future improvements, such as incorporating automated hyperparameter tuning methods or meta-optimization frameworks.
>
> ## 3. Evaluations with Additional Models:
> While our methods are inherently data-centric, we agree that demonstrating their transferability across different detectors strengthens our findings. We implement YOLOv3 as an additional detector and apply RCC and RCF on KITTI to evaluate adaptability. This serves as a proof of concept, validating that our proposed methods generalize to other detectors without requiring modifications. The results of this experiment will be included in the ablation studies (Section 4.5) of the revised paper.
>
> ## 4. Clarification on Pseudo-Label Selection (PLS):
> We appreciate your observation regarding the terminology used to describe PLS. Referring to it as a "metric" may indeed cause confusion. We will revise the paper to consistently describe PLS as a method or building block. Instead, we will use $D_i$ or $S_i$ explicitly to refer to the metrics introduced as part of the PLS method. This clarification enhances the precision and readability of the paper.
>
> ## Conclusion:
> Thank you again for your thoughtful and constructive review. We hope that these changes and additional experiments address your concerns and further emphasize the robustness, usability, and generalizability of our methods. We will upload a revised version of the paper incorporating these changes and new results as soon as the ongoing experiments are completed.

---

### Comment · Reviewer_2jgj · 2025-01-20
**reversion**

I am satisfied with the changes that being made.

I still suggest to move some of the results, ablations, collages examples and some of the elaborated tables in 4.3 (the text of page 11) to an appendix.

It's mostly very important to the story and validation of your work, but as now it interrupting the paper from being a coherent story.
Maybe it's only my opinion, but i would be happier with a paper that is ~12-13 pages long and rest of the information is in the appendix.


also most tables can be smaller

---

> ### Author Response · Authors · 2025-01-31
>
> We sincerely appreciate the active discussion and your valuable feedback throughout the review process. We are pleased to hear that you are satisfied with the changes we have made.
>
> Regarding your suggestion to move some content to an appendix, we carefully considered this when preparing our submission. One of the reasons we explicitly chose TMLR was to avoid strict page limits, allowing us to keep all relevant content in the main paper without requiring readers to jump between sections and an appendix. Given the interconnected nature of our analysis, we believe that presenting everything in one place helps maintain a coherent narrative and ensures that all methodological details and findings remain directly accessible.
>
> That said, we understand that this is a subjective preference, and we appreciate your perspective on improving readability. If other reviewers also request this change, or if you strongly advocate for it, we will be happy to reconsider and adjust accordingly.
>
> Thank you again for your thoughtful feedback and for engaging in this discussion. We truly appreciate your time and valuable insights.

---

### Decision · Action_Editor_Zukd · 2025-03-12

**Recommendation:** Accept as is

**Comment:**

The paper presents a novel empirical study addressing challenges in semi-supervised object detection (SSOD) for self-driving datasets, focusing on mitigating class imbalance, ground-truth label noise, and pseudo-label inaccuracies. The reviewers unanimously recommend acceptance, commending the work's clarity, practical interventions, and comprehensive experiments. Key strengths include the logical organization of content, effective visual aids, and actionable insights for the community. While initial concerns were raised about the scope of claims and the generality of data/model-agnosticism, the authors addressed these through revisions, as noted in the reviews.

Therefore, I recommend accepting this paper.

**Audience:**

Yes, the findings of this paper would likely interest a significant portion of TMLR’s audience whose research interests including semi-supervised learning and object detection.

**Claims And Evidence:**

Yes. The claims made in the submission are supported by the empirical results.